# Peptidoglycan recycling is critical for cell division, cell wall integrity, and β-lactam resistance in *Caulobacter crescentus*

Pia Richter[1], Anna Merz[1], Jacob Biboy[2], Nicole Paczia[3], Timo Glatter[4], Jared Ng[5], Waldemar Vollmer[5], Martin Thanbichler[1,6,7]*

[1]Department of Biology, Marburg University, Marburg, Germany; [2]Centre for Bacterial Cell Biology, Institute for Cell and Molecular Biosciences, Newcastle University, Newcastle upon Tyne, United Kingdom; [3]Core facility for Metabolomics and Small-Molecule Mass Spectrometry, Max Planck Institute for Terrestrial Microbiology, Marburg, Germany; [4]Core facility for Mass Spectrometry and Proteomics, Max Planck Institute for Terrestrial Microbiology, Marburg, Germany; [5]Institute for Molecular Bioscience, University of Queensland, Brisbane, Australia; [6]Max Planck Fellow Group Bacterial Cell Biology, Max Planck Institute for Terrestrial Microbiology, Marburg, Germany; [7]Center for Synthetic Microbiology (SYNMIKRO), Marburg, Germany

## eLife Assessment

This manuscript presents a **valuable** investigation of the peptidoglycan (PG) recycling pathway in *Caulobacter crescentus*. The authors showed that PG recycling in *C. crescentus* is essential not only for β-lactam (ampicillin) resistance but also for cell morphology, efficient division, and overall fitness. The study is comprehensive and **compelling**.

*For correspondence:
thanbichler@uni-marburg.de

**Abstract** Most bacteria possess a peptidoglycan (PG) sacculus, which is continuously remodeled during cell growth and division. The turnover products generated in this process are typically imported into the cell and reused for PG biosynthesis. While the underlying pathways have been studied intensively in gammaproteobacteria, knowledge of their presence and physiological roles in other bacterial lineages remains limited. Here, we comprehensively investigate PG recycling in the alphaproteobacterial model organism *Caulobacter crescentus*. Characterizing the activities of key enzymes in vitro and in vivo, we show that this species contains a functional PG recycling pathway, including the MurU shunt. Our results reveal that PG recycling is critical for *C. crescentus* cell morphology and division and is dynamically regulated to balance the flux of metabolic intermediates toward PG biosynthesis and central carbon metabolism. Importantly, defects in PG recycling strongly impair the intrinsic ampicillin resistance of *C. crescentus* without changing the activity of its β-lactamase BlaA, likely by limiting PG precursor biosynthesis and thereby decreasing the activity of the cell wall biosynthetic machinery in the presence of residual antibiotic. These findings underscore the central role of PG recycling in bacterial fitness and suggest that inhibiting this process could provide a promising strategy to combat β-lactam-resistant pathogens.

## Introduction

In most bacteria, the cytoplasmic membrane is surrounded by a layer of peptidoglycan (PG), which protects the cell from osmotic lysis, maintains cell shape, and serves as a physical barrier against

**Figure 1.** The *C. crescentus* genome encodes a complete peptidoglycan (PG) recycling pathway. De novo PG biosynthesis starts with the conversion of fructose-6-phosphate (Fru-6-P), supplied by central carbon metabolism, to glucosamine-6-phosphate (GlcN-6-P). This key metabolite is then transformed by multiple enzymatic steps into the membrane-attached PG precursor lipid II, which is flipped to the periplasmic face and incorporated into the existing PG sacculus by glycosyltransferases (GTases) and transpeptidases (TPases). To enable PG remodeling, growth, and cell separation, the PG meshwork is cleaved by a diverse set of enzymes. The anhydro-muropeptides and *N*-acetylglucosamine (GlcNAc) molecules generated in this process are imported into the cytoplasm by the permease AmpG or two homologs of the phosphotransferase system (PTS) transporter NagE, respectively. In the cytoplasm, the peptide is released by AmiR and the sugars are separated by NagZ. Afterwards, two independent pathways feed the two sugars back into the de novo PG biosynthetic pathway. The enzymes are named as established for *E. coli* and *P. aeruginosa*, with the exception of the *C. crescentus* homolog of AmpD, which was renamed to AmiR due to its different domain organization. The enzymes analyzed in this study are indicated in bold type. Symbols are defined in the legend on the right.

environmental assaults (*Egan et al., 2020*). PG is composed of a dense meshwork of glycan strands made of alternating *N*-acetylglucosamine (GlcNAc) and *N*-acetylmuramic acid (MurNAc) moieties that are connected by short peptides containing unusual D-amino acids (*Vollmer et al., 2008b*). The integrity of the PG layer is critical for bacterial fitness and survival, which makes it a major target of medically relevant antibiotics (*Bugg et al., 2011*).

PG biosynthesis initiates in the cytoplasm (*Barreteau et al., 2008*; *Höltje, 1998*), starting with the transformation of fructose-6-phosphate (Fru-6-P) to UDP-MurNAc. Next, this key intermediate is modified with a tripeptide, synthesized stepwise by dedicated amino acid ligases, and completed by the addition of a preformed D-Ala–D-Ala unit, leading to UDP-MurNAc-pentapeptide (*Figure 1*).

The MurNAc-pentapeptide moiety is then transferred to the lipid carrier undecaprenol phosphate and linked to GlcNAc. Subsequently, the product, lipid II, is flipped across the cytoplasmic membrane (*Sham et al., 2014*), where it is used by glycosyltransferase (GTases) to polymerize the glycan chains. Transpeptidases (TPases) finally crosslink the peptides of neighboring glycan strands, thereby establishing the mesh-like architecture of PG (*Egan et al., 2020*; *Höltje, 1998*). The most abundant group of TPases are the DD-TPases, which cleave the C-terminal D-Ala–D-Ala unit of a pentapeptide and then transfer the resulting tetrapeptide to an adjacent acceptor peptide (*Sauvage et al., 2008*). This group of enzymes is also known as penicillin-binding proteins (PBPs), because they recognize penicillin and other β-lactams as non-cognate substrates, based on the structural similarity of their β-lactam ring to the D-Ala–D-Ala moiety of uncrosslinked peptides. These compounds form a stable adduct with the active-site serine of PBPs and thus irreversibly inhibit their crosslinking activity, thereby weakening the cell wall and inducing cell lysis (*Cho et al., 2014*; *Mora-Ochomogo and Lohans, 2021*). The sensitivity of DD-TPases to β-lactams represents a vulnerability that is widely exploited to treat bacterial infections, but resistance to this class of antibiotics is rapidly spreading among pathogens (*Lima et al., 2020*).

Cells usually contain both monofunctional PBPs, which contain only a transpeptidase domain, and bifunctional PBPs, which contain an additional GTase domain and thus have polymerization as well as crosslinking activity (*Sauvage et al., 2008*; *Vollmer and Bertsche, 2008a*). To enable the tight spatiotemporal control of PG biosynthesis, these enzymes are often part of dynamic multi-protein complexes that include regulatory factors and cytoskeletal elements (*Egan et al., 2020*; *Typas et al., 2012*). In most rod-shaped bacteria, lateral cell wall elongation is carried out by the elongasome, a complex organized by the actin-like protein MreB (*Jones et al., 2001*). Multiple elongasomes are distributed along the longitudinal axis of the cell and move around its circumference in helical tracks (*Garner et al., 2011*; *van Teeffelen et al., 2011*), incorporating new cell wall material in a dispersed manner. Their PG biosynthetic activity is mediated by the GTase RodA and the monofunctional DD-TPase PBP2 (*Cho et al., 2016*; *Meeske et al., 2016*), whose functions are closely coupled (*Rohs et al., 2018*). Cell division, by contrast, is mediated by the divisome, which is orchestrated by polymers of the tubulin homolog FtsZ (*Bi and Lutkenhaus, 1991*; *Bisson-Filho et al., 2017*; *Yang et al., 2017*), with the GTase FtsW and the monofunctional DD-TPase PBP3 acting as the main PG biosynthetic enzymes (*Taguchi et al., 2019*). The division process also involves bifunctional PBPs, such as PBP1B in *E. coli* (*Bertsche et al., 2006*; *Navarro et al., 2025*), although this class of enzymes also appears to act outside larger complexes in cell wall repair (*Morè et al., 2019*; *Pazos and Vollmer, 2021*).

To enable the incorporation of new cell wall material into the existing sacculus, the PG meshwork needs to be opened at specific sites – a task that is achieved by a diverse set of autolysins (*van Heijenoort, 2011*; *Vollmer et al., 2008c*). DD- and LD-endopeptidases cleave the peptides and the crosslinks between them, while amidases remove the entire peptide stem from the glycan chain. The glycan chains are cleaved by glucosaminidases and muramidases. The latter include lytic transglycosylases, which break the β–1,4-glycosidic bond between MurNAc and GlcNAc, leading to the concomitant formation of a 1,6-anhydro bond in the MurNAc moiety (*Dik et al., 2017*; *Vollmer et al., 2008c*). The activity of autolytic enzymes generates small PG fragments, called muropeptides. Only a small fraction of these is released into the environment (*Goodell and Schwarz, 1985*), whereas the majority is transported into the cytoplasm, where they are metabolized and reused in PG synthesis. This process is known as PG recycling and has so far been studied mainly in gammaproteobacteria, such as *Escherichia coli* or *Pseudomonas aeruginosa* (*Dik et al., 2018*; *Gilmore and Cava, 2025*; *Park and Uehara, 2008*).

During one division cycle, more than 50% of the PG sacculus is broken down in *E. coli*. Most of the turnover products in this and other Gram-negative species are 1,6-anhydro-muropeptides (*Vollmer et al., 2008c*). They are transported into the cytoplasm by the proton-driven Major Facilitator Superfamily (MFS) permease AmpG (*Cheng and Park, 2002*; *Jacobs et al., 1994*), where the sugars and the peptide are separated from each by the combined action of the LD-carboxypeptidase LdcA (*Templin et al., 1999*), the amidase AmpD (*Höltje et al., 1994*), and the β-hexosaminidase NagZ (*Cheng et al., 2000*). The GlcNAc moiety is then recycled by conversion to GlcNAc-6-P and then glucosamine-6-P (GlcN-6-P), which can either be transformed into the PG precursor UDP-MurNAc or directed into central carbon metabolism by deamination to Fru-6-P (*Gilmore and Cava, 2025*; *Vollmer et al., 2008c*). The fate of the 1,6-anhydro-MurNAc (anhMurNAc) moiety, on the other hand, varies between

species. In *E. coli*, it is converted to GlcNAc-6-P (*Uehara et al., 2005*), whereas *P. aeruginosa* employs the MurU pathway to convert anhMurNAc into UDP-MurNAc, thereby channeling it directly back into PG biosynthesis (*Borisova et al., 2017*; *Fumeaux and Bernhardt, 2017*; *Gisin et al., 2013*; *Figure 1*). As the last component, the peptide is typically broken down to single amino acids that are subsequently reused to synthesize new peptide stems (*Gilmore and Cava, 2025*). An alternative route has been reported for *E. coli*, which trims the released peptide by removal of the terminal D-Ala residues and then attaches the resulting tripeptide to UDP-MurNAc using the peptide ligase Mpl (*Mengin-Lecreulx et al., 1996*).

The recycling of PG degradation products is an efficient way to save energy and resources, which may be particularly relevant during growth in nutrient-limiting conditions (*Gilmore and Cava, 2025*). For human commensals or pathogens, it also represents an important strategy for evading detection by the innate immune system, which senses released muropeptides through the NOD1 and NOD2 receptors (*Trindade and Chen, 2020*). Apart from that, some gammaproteobacteria, such as *Citrobacter freundii* and *P. aeruginosa*, use PG recycling as a sensory mechanism to detect the presence of β-lactam antibiotics and initiate a resistance response (*Dik et al., 2018*). These species possess the inducible β-lactamase AmpC, whose gene is controlled by the LysR-type transcriptional regulator AmpR. The activity of AmpR is modulated through its interaction with muropeptides (*Jacobs et al., 1994*; *Lindquist et al., 1989*). In the absence of β-lactams, it is associated with UDP-MurNAc-pentapeptide, acting as a repressor that prevents *ampC* expression (*Jacobs et al., 1997*). When cells are exposed to β-lactam antibiotics, the disruption of PG metabolism leads to an increased accumulation of anhydro-muropeptides, thus raising the levels of PG recycling intermediates in the cytoplasm. Some of these intermediates, especially anhMurNAc-pentapeptide (*Dietz et al., 1997*; *Vadlamani et al., 2015*), can interact with AmpR and competitively displace its co-repressor UDP-MurNAc-pentapeptide. As a result, AmpR transforms into a transcriptional activator that stimulates *ampC* expression, thereby establishing β-lactam resistance (*Dietz et al., 1997*; *Dik et al., 2018*; *Jacobs et al., 1997*). While PG recycling and its impact on cell physiology as well as β-lactam resistance have been extensively studied in gammaproteobacteria, these processes are still poorly understood in other bacterial lineages.

Here, we studied PG recycling in the alphaproteobacterium *Caulobacter crescentus*, a species intrinsically resistant to β-lactam antibiotics due to production of the β-lactamase BlaA, extending recent work on this topic by *Modi et al., 2025*. We found that *C. crescentus* encodes a functional PG recycling pathway and characterized the activities of key enzymes, using a combination of biochemical, cell biological, metabolomics, and proteomics analyses. Our results demonstrate that PG recycling is required to sustain proper cell morphology and division in this species, especially upon transition to stationary phase. They further show that *C. crescentus* cells regulate their PG recycling pathway to control the flux of internal and external GlcNAc into PG biosynthesis and central carbon metabolism, while employing the MurU pathway to directly convert anhMurNAc back into the PG precursors. Importantly, PG recycling-deficient cells show severe defects in β-lactam resistance, despite normal BlaA production and activity. For the model β-lactam ampicillin, this phenotype can be explained by increased sensitivity of the septal TPase PBP3 to ampicillin-mediated inhibition, likely resulting from a reduced supply of PG building blocks in the absence of PG recycling. Together, our findings indicate that β-lactam efficacy is modulated by the availability of PBP substrates, uncovering a new aspect of the complex interplay between this major class of antibiotics and their cellular targets.

## Results

### The genome of *C. crescentus* encodes a full PG recycling pathway

To initiate the analysis of PG recycling in *C. crescentus*, we first mined the genome of this species for homologs of proteins that are central to the well-studied PG recycling pathway of *E. coli*. Reciprocal BLAST analysis revealed clear counterparts for the muropeptide permease AmpG as well as for most enzymes involved in the processing of the imported anhydro-muropeptides and the transformation of their sugar and peptide moieties into new PG building blocks (*Table 1* and *Figure 1*). Notably, no homologs were found for the LD-carboxypeptidase LdcA and the peptide ligase Mpl, which trim tetrapeptides to tripeptides and attach these to UDP-MurNAc, respectively. Another key enzyme missing in *C. crescentus* is the etherase MurQ, a critical player in the recycling of anhMurNAc,

**Table 1.** List of peptidoglycan recycling enzymes found in *E. coli* and *P. aeruginosa* and the corresponding homologs of *C. crescentus* identified by reciprocal BLAST analysis.

| Protein | Homolog(s) | Sequence identity (%) | E value | Function |
|---|---|---|---|---|
| *E. coli** | *C. crescentus* | | | |
| AmpG | CCNA_00136 | 40.4 | 3E-40 | Muropeptide permease |
| NagE | CCNA_00572<br>CCNA_00458 | 45.0<br>45.0 | 2E-131<br>3E-131 | *N*-acetylglucosamine-specific IIBC component (PTS system) |
| LdcA | - | - | - | LD-carboxypeptidase |
| AmpD | CCNA_02650 | 33.5 | 5E-20 | 1,6-anhydro-*N*-acetylmuramyl-L-alanine amidase |
| NagZ | CCNA_02085 | 34.7 | 6E-44 | β-*N*-acetyl-D-glucosaminidase |
| NagK | CCNA_03849 | 41.1 | 2E-66 | *N*-acetyl-D-glucosamine kinase |
| AnmK | CCNA_01945 | 34.6 | 2E-49 | anhydro-*N*-acetylmuramic acid kinase |
| MurQ | - | - | - | *N*-acetylmuramic acid-6-phosphate etherase |
| NagA | CCNA_00568<br>CCNA_00452 | 36.3<br>39.5 | 6E-76<br>1E-84 | *N*-acetylglucosamine-6-phosphate deacetylase |
| GlmM | CCNA_00116 | 51.5 | 4E-153 | Phosphoglucosamine mutase |
| GlmU | CCNA_02389 | 40.4 | 2E-97 | UDP-*N*-acetylglucosamine pyrophosphorylase |
| *P. aeruginosa†* | *C. crescentus* | | | |
| MupP | CCNA_02390 | 32.6 | 3E-27 | Phosphoglycolate phosphatase |
| AmgK | CCNA_03649 | 30.1 | 1E-32 | *N*-acetylmuramate/*N*-acetylglucosamine kinase |
| MurU | CCNA_03650 | 36.6 | 4E-26 | *N*-acetylmuramate-α–1-phosphate uridylyltransferase |

* strain *E. coli* K-12 substr. MG1655 (Genbank accession code: U00096.3).
† strain *P. aeruginosa* PAO1 (Genbank accession code: AE004091.2).

converting MurNAc-6-P into GlcNAc-6-P and thus enabling its re-utilization through the GlcNAc recycling/degradation pathway (*Jaeger et al., 2005*; *Uehara et al., 2006*). Instead, we identified a three-step pathway initially found in *P. aeruginosa* (*Borisova et al., 2017*; *Fumeaux and Bernhardt, 2017*; *Gisin et al., 2013*) that leaves the MurNAc moiety intact and uses it directly to regenerate UDP-MurNAc. These findings suggest that the two sugar moieties of anhydro-muropeptides are recycled independently in *C. crescentus*. The protein homologs identified are generally well-conserved at the sequence level (*Table 1*) and show the same domain architectures as their *E. coli* and *P. aeruginosa* counterparts. The only exception is the homolog of AmpD, in which the amidase_2 domain is fused to a C-terminal PG_binding_1 domain that is not present in the *E. coli* protein. Due to this difference, this protein will be referred to as 'AmiR' (*Amid*ase for PG *R*ecycling) throughout the present study.

## Predicted key players in the PG recycling pathway are functionally active

To test the prediction that *C. crescentus* can recycle PG degradation products, we aimed to verify the activity of the two first enzymes in the recycling pathway, AmiR and NagZ, which mediate the release of the peptide and sugar moieties, thereby providing the basis for their refunneling into PG biosynthesis (see *Figure 1*). To this end, pentapeptide-rich PG sacculi isolated from a DD-carboxypeptidase-deficient *E. coli* strain were treated with the lytic transglycosylase Slt from *E. coli* (*Betzner and Keck, 1989*) to release anhydro-muropeptides (*Figure 2A*). These were then incubated with AmiR or a catalytically inactive variant of the protein (AmiR*) in which three conserved amino acid residues

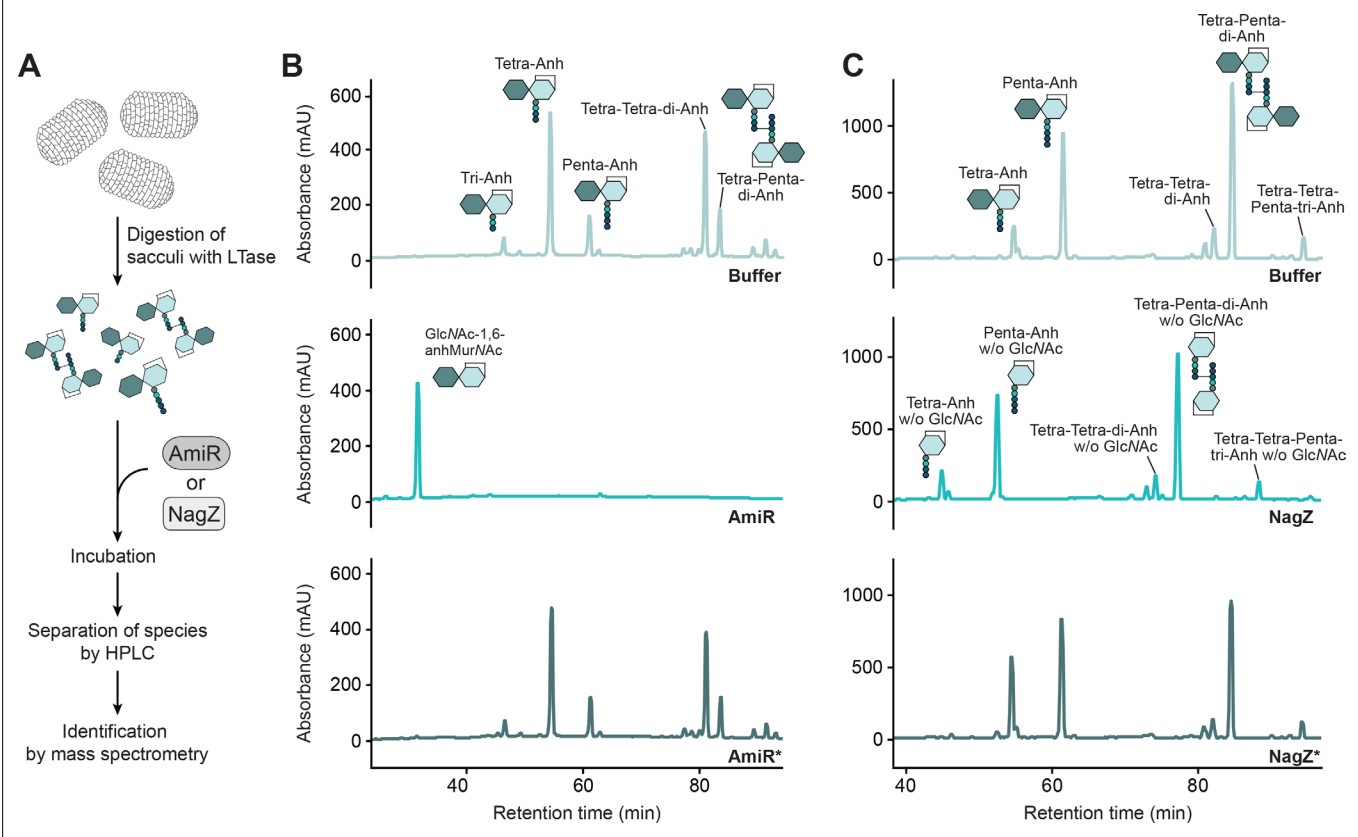

**Figure 2.** *C. crescentus* AmiR and NagZ are functional and have a broad substrate specificity. (**A**) Overview of the peptidoglycan (PG) digestion assay used to assess the hydrolytic activities of AmiR and NagZ. (**B**) HPLC chromatograms showing the products generated upon incubation of anhydro-muropeptides from lytic transglycosylase-treated PG sacculi without added proteins (Buffer), with AmiR or with a catalytically inactive AmiR variant (AmiR*). The identities of anhMurNAc-peptides were determined based on their known retention times (*Glauner, 1988*). The AmiR reaction product GlcNAc–1,6-anhMurNAc was identified by mass spectrometry (*Figure 2—figure supplement 4A*). (**C**) HPLC chromatograms showing the products generated upon incubation of anhydro-muropeptides from lytic transglycosylase-treated PG sacculi without added proteins (Buffer), with NagZ or with a catalytically inactive NagZ variant (NagZ*). Disaccharide-containing anhydro-muropeptides were identified based on their known retention times (*Glauner, 1988*). The identities of anhMurNAc-peptides were validated by mass spectrometry (*Figure 2—figure supplement 4B*). Note that the PG sacculi used in the AmiR and NagZ activity assays were isolated from different *E. coli* strains that exhibit distinct muropeptide profiles (see Methods).

The online version of this article includes the following figure supplement(s) for figure 2:

**Figure supplement 1.** Substrate specificity of AmiR.

**Figure supplement 2.** Lack of AmiR activity towards intact peptidoglycan (PG) sacculi.

**Figure supplement 3.** Activity of NagZ with muropeptides.

**Figure supplement 4.** Mass spectrometric identification of the AmiR and NagZ reaction products.

**Figure supplement 5.** Activity of AmiR against anhydro-*N*-acetylmuramic acid (anhMurNAc)-tetrapeptide.

coordinating the essential $Zn^{2+}$ cofactor of AmpD-like amidases (*Généreux et al., 2004*; *Liepinsh et al., 2003*; *Uehara and Park, 2007*) were replaced by alanine (*Figure 2—figure supplement 1A*). HPLC analysis of the reaction products revealed that AmiR was able to remove the peptide stems from all detectable anhydro-muropeptides, regardless of the length of the peptide or the number of cross-linked peptides (*Figure 2B*). By contrast, no amidase activity was detected in the control reactions containing the mutant AmiR* protein, confirming the critical role of the metal cofactor in the amidase reaction. To determine whether, despite its broad substrate specificity, AmiR had a preference for certain anhydro-muropeptides, we repeated the enzyme assay with limiting enzyme concentrations. The results showed that AmiR efficiently hydrolyzed tetrapeptide-containing fragments, whereas anhydro-muropeptides with tripeptide or pentapeptide stems were cleaved with lower efficiency (*Figure 2—figure supplement 1B*). Notably, unlike *E. coli* AmpD (*Höltje et al., 1994*), it also showed activity towards muropeptides that were produced with the muramidase cellosyl and thus contained

reducing Mur*N*Ac instead of anhMur*N*Ac moieties, although the efficiencies were slightly lower under these conditions (*Figure 2—figure supplement 1C*). By contrast, no activity was detectable with undigested peptidoglycan sacculi, consistent with a role of the protein in the recycling of PG fragments (*Figure 2—figure supplement 2*). These results confirm that AmiR functions as a *bona fide* amidase involved in peptidoglycan recycling.

Next, we used the same approach to study the biochemical activity of *C. crescentus* NagZ. For this purpose, anhydro-muropeptides or reducing muropeptides were incubated with wild-type NagZ or a catalytically inactive variant (NagZ*) in which an essential aspartic acid residue at the catalytic site (*Bacik et al., 2012*) was replaced with an alanine (*Figure 2—figure supplement 3A*). NagZ removed the Glc*N*Ac moiety from both types of muropeptides and, in each case, hydrolyzed all detectable muropeptide variants, irrespective of the length of the peptide or the degree of crosslinkage (*Figure 2C* and *Figure 2—figure supplement 3B*). By contrast, NagZ* did not show any activity. Thus, similar to its *E. coli* homolog (*Cheng et al., 2000*), NagZ acts as an *N*-acetyl-β-D-glucosaminidase with broad substrate specificity. Together, these findings demonstrate that *C. crescentus* possesses enzymes catalyzing the two initial steps of PG recycling.

Previous work has shown that the reactions catalyzed by *C. freundii* AmpD and NagZ do not constitute a strictly linear pathway but can occur in either order, with AmpD also acting on anhydro-muropeptides that were pretreated with NagZ and thus lack a Glc*N*Ac residue (*Jacobs et al., 1995*). To determine whether this was also the case for *C. crescentus* AmiR, we specifically assessed its in vitro activity toward the substrate anhMur*N*Ac-tetrapeptide, generated by preincubating purified Glc*N*Ac–anhMur*N*Ac-tetrapeptide with NagZ. Mass spectrometric analysis revealed that AmiR-containing reactions accumulated substantial amounts of free anhMur*N*Ac and free tetrapeptide, while only background levels of this product were observed in reactions containing AmiR* or lacking added protein (*Figure 2—figure supplement 5*). These results demonstrate that the order of the two initial PG recycling steps is also interchangeable in *C. crescentus*.

In *E. coli*, PG turnover products are imported by various transporters with different binding specificities (*Gilmore and Cava, 2025*), but only AmpG recognizes muropeptides containing a Glc*N*Ac–anhMur*N*Ac unit, as typically produced by lytic transglycosylases (*Chang et al., 2025*; *Cheng and Park, 2002*; *Jacobs et al., 1994*). To clarify whether the AmpG homolog of *C. crescentus* has a similar role, we used a metabolomics approach to detect imported anhydro-muropeptides, using the NagZ

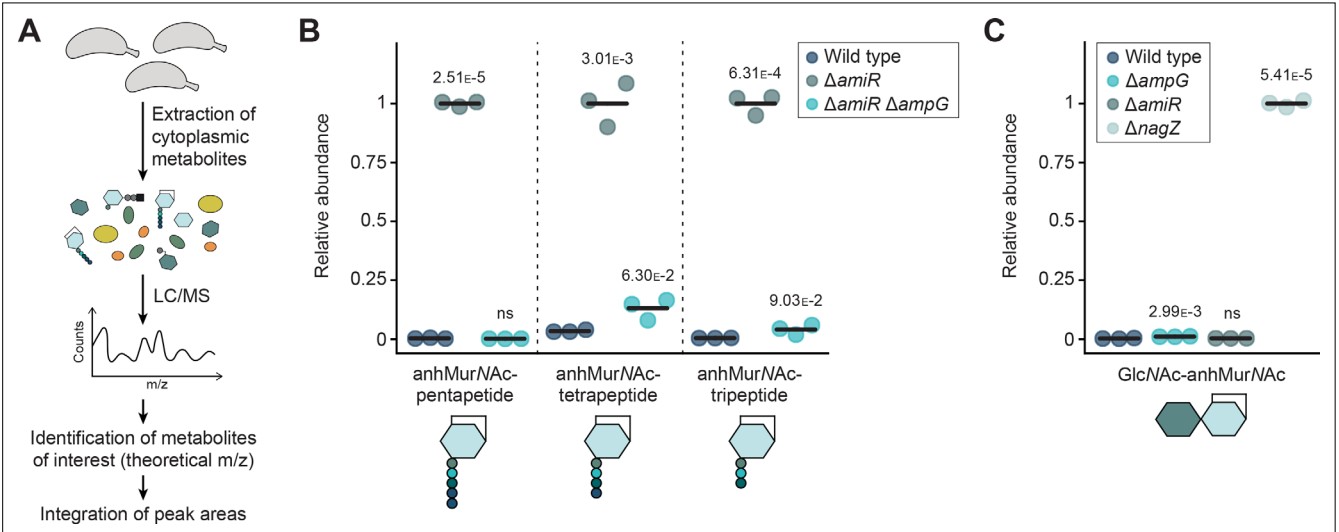

**Figure 3.** Metabolomic analysis reveals aberrant levels of peptidoglycan (PG) recycling intermediates in PG recycling-deficient strains. (**A**) Overview of the analysis pipeline used to identify cytoplasmic PG recycling products. (**B**) Relative levels of the indicated anhMur*N*Ac-peptide species in the cytoplasm of *C. crescentus* wild-type, Δ*amiR* (PR033) and Δ*amiR* Δ*ampG* (PR221) cells, measured by metabolomics analysis. For each anhydro-muropeptide, the mass spectrometric peak areas were normalized against the mean obtained for the Δ*amiR* mutant. The statistical significance of differences between the wild-type and the mutant strains was determined using a two-sided Welch's *t*-test. ns indicates *p*-values >0.1. (**C**) Relative levels of the Glc*N*Ac–anhMur*N*Ac disaccharide in *C. crescentus* wild-type, Δ*ampG* (PR207), Δ*amiR* (PR033), and Δ*nagZ* (PR188) cells, assessed as for panel B. The mass spectrometric peak areas were normalized against the mean obtained for the Δ*nagZ* mutant. Statistical significance was determined as described for panel B.

cleavage products anhMur*N*Ac-pentapeptide, -tetrapeptide, and -tripeptide as a proxy (*Figure 3A*). In the wild-type strain, the levels of these PG fragments were close to the detection limit, suggesting that they are rapidly processed into smaller units. Consistent with this notion, and in line with previous results (*Jacobs et al., 1994*; *Simpson et al., 2023*), a strong accumulation of these species was observed in a Δ*amiR* mutant, which lacks the amidase to remove the peptide stem (*Figure 3B*). Upon the deletion of *ampG* in the Δ*amiR* background, the three anhydro-muropeptides were again barely detectable, confirming that AmpG functions as the primary importer of lytic transglycosylase-generated PG degradation products in *C. crescentus*.

The analysis of cytoplasmic PG degradation intermediates also provided a means to obtain further insight into the nature and relevance of the reactions catalyzed by AmiR and NagZ. Consistent with the results of the in vitro activity assays (compare *Figure 2C*), the accumulation of anhMur*N*Ac-peptide species in the absence of AmiR (*Figure 3B*) shows that NagZ can hydrolyze the glycosidic bond in the Glc*N*Ac–anhMur*N*Ac unit while the peptide stem is still attached. Moreover, it indicates that the reaction catalyzed by AmiR is a bottleneck in the PG recycling pathway, because the removal of the peptide appears to be critical for downstream reactions to occur. To clarify whether cleavage of the Glc*N*Ac–anhMur*N*Ac bond was required to further process the disaccharide unit after peptide removal, we assessed the levels of the free disaccharide in different mutant backgrounds. Metabolomics analysis revealed that Glc*N*Ac–anhMur*N*Ac was barely detectable in the wild-type strain, suggesting that the initial decomposition of anhydro-muropeptides by NagZ and AmiR is highly efficient. The disaccharide was also not enriched in Δ*ampG* and Δ*amiR* cells, but highly accumulated in the Δ*nagZ* mutant (*Figure 3C*). Thus, the two sugars need to be separated before further processing steps can initiate.

Together, our analyses verify that *C. crescentus* possesses the machinery required to import anhydro-muropeptides and break them down into their sugar and peptide components. Moreover, they indicate that AmiR and NagZ have a broad substrate specificity, so that the initial steps of the recycling process do not occur in a strictly linear fashion.

## PG recycling is necessary for proper cell division and β-lactam resistance in *C. crescentus*

Although a considerable fraction of the PG sacculus (~50% in *E. coli*) is turned over and recycled during each cell division cycle (*Goodell, 1985*; *Goodell and Schwarz, 1985*), the model species *E. coli* and *P. aeruginosa* typically do not display any obvious morphological phenotype upon disruption of the PG recycling pathway (*Park and Uehara, 2008*; *Gilmore and Cava, 2025*). The only exception are *E. coli* cells with a defect in the LD-carboxypeptidase LdcA, which promotes the erroneous ligation of UDP-Mur*N*Ac with unprocessed tetrapeptides, blocking the addition of the terminal D-Ala–D-Ala unit and inducing cell lysis in the stationary growth phase (*Templin et al., 1999*). To determine the relevance of PG recycling in *C. crescentus*, we examined the morphology of Δ*ampG*, Δ*amiR,* and Δ*nagZ* cells growing in rich (PYE) medium by light microscopy. In each case, the mutation caused significant cell filamentation and widening, with the defects becoming more pronounced upon transition of the cells from the exponential to the stationary growth phase (*Figure 4A and B*). These phenotypes were reversed by ectopic expression of the deleted genes under the control of a xylose-inducible promoter, even when cells were grown in the absence of inducer (*Figure 4—figure supplement 1A and B*), suggesting that low levels of these proteins resulting from basal promoter activity are sufficient for proper function. Strains in which the *amiR* and *nagZ* genes were replaced with alleles encoding the catalytically inactive AmiR* or NagZ* showed phenotypes similar to those of the respective deletion mutants, verifying that the defects observed were due to the absence of the enzymatic activities (*Figure 4—figure supplement 2*). Together, these results indicate that PG recycling is critical for proper cell division and, to some extent, also lateral growth in *C. crescentus*. Notably, the lack of AmpG or AmiR, which prevents the recycling of the peptide stem (*Figure 3B*), caused a more severe phenotype than the lack of NagZ, which blocks the processing of the free Glc*N*Ac–anhMur*N*Ac-disaccharide (*Figure 3C*). Therefore, the supply of amino acids rather than sugars appears to be the major limiting factor in PG precursor biosynthesis. To further investigate the effect of nutrients on the phenotypes of the different mutant strains, we also analyzed cells grown to stationary phase in minimal (M2G) medium. Interestingly, none of the mutant strains showed obvious morphological defects under these conditions (*Figure 4—figure supplement 3A and B*), possibly because the high

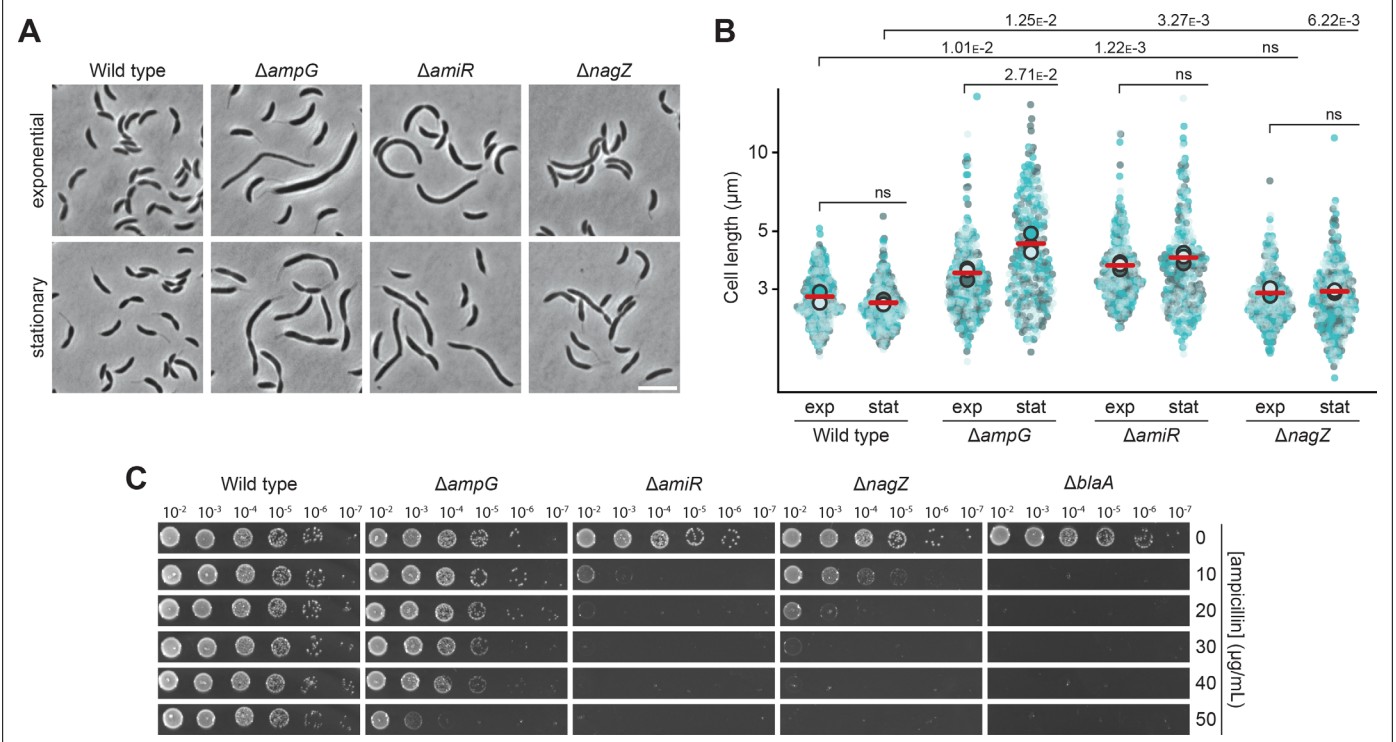

**Figure 4.** AmpG, AmiR, and NagZ are required for proper cell shape and β-lactam resistance in *C. crescentus*. (**A**) Phase contrast images of *C. crescentus* wild-type, Δ*ampG* (PR207), Δ*amiR* (PR033), and Δ*nagZ* (PR188) cells harvested in the exponential and stationary growth phase. Scale bar: 5 μm. (**B**) Superplots showing the distribution of cell lengths in populations of *C. crescentus* wild-type, Δ*ampG* (PR207), Δ*amiR* (PR033), and Δ*nagZ* (PR188) cells in the exponential (exp) and stationary (stat) growth phase. Small dots represent the data of three independent replicates (shown in light blue, teal, and dark gray; n=100 cells per replicate). Large dots represent the mean values of the three datasets, with their average indicated by a red horizontal line. The statistical significance (*p*-value) of differences between conditions was assessed using a two-sided Welch's *t*-test. ns indicates *p*-values>0.1. (**C**) Serial dilution spot assay investigating the growth of wild-type, Δ*ampG* (PR207), Δ*amiR* (PR033), Δ*nagZ* (PR188), and Δ*blaA* (CS606) cells on agar plates with different concentrations of ampicillin. See *Source data 1* for all replicates (n=3 independent experiments).

The online version of this article includes the following figure supplement(s) for figure 4:

**Figure supplement 1.** Reversion of the phenotypes of peptidoglycan recycling-deficient strains through complementation with the corresponding wild-type genes.

**Figure supplement 2.** Morphological characterization of catalytically inactive AmiR and NagZ variants.

**Figure supplement 3.** Growth of peptidoglycan recycling-deficient mutants in minimal medium.

**Figure supplement 4.** Redundant role of the soluble lytic transglycosylase SdpA in the production of anhydro-muropeptides.

glucose content of M2G medium compensates for the lack of PG recycling by enhancing de novo PG synthesis.

Apart from promoting growth, PG recycling was found to critically contribute to the regulation of β-lactam resistance in organisms, such as *C. freundii* and *P. aeruginosa*, which use the level of anhMur*N*Ac-pentapeptide as a readout to sense the presence of β-lactams and control the production of the β-lactamase AmpC (***Dietz et al., 1997***; ***Dik et al., 2018***; ***Jacobs et al., 1997***). Previous work has also suggested a link between PG recycling and β-lactam resistance in *C. crescentus* (***Modi et al., 2025***), a species that produces a metallo-β-lactamase (BlaA) conferring intrinsic high-level resistance to ampicillin and related β-lactam antibiotics (***Docquier et al., 2002***; ***West et al., 2002***). To further clarify the underlying mechanism, we analyzed the ability of the *C. crescentus* Δ*ampG*, Δ*amiR*, and Δ*nagZ* mutants to grow on nutrient-agar (PYE) plates at different ampicillin concentrations (***Figure 4C***). Wild-type cells tolerated high levels (50 μg/mL) of the antibiotic without any apparent fitness defects. In comparison, all three mutant strains showed a significant reduction in ampicillin resistance. The strongest effect was observed for Δ*amiR* cells, whose sensitivity approached that of Δ*blaA* cells (***Figure 4C***). Surprisingly, the Δ*ampG* mutant showed the least pronounced resistance defect, although it exhibited the largest fraction of filamentous cells (compare ***Figure 4A and B***),

indicating that the defects in β-lactam resistance and cell morphology are not strictly connected. The mutant phenotypes were largely reversed by ectopic expression of the deleted genes from an inducible promoter, even when cells were analyzed in the absence of the inducer (*Figure 4—figure supplement 1C*), thereby verifying the validity of the deletion analysis. Interestingly, while growth in minimal (M2G) medium largely abolished the cell filamentation phenotype of PG recycling-deficient cells (compare *Figure 4—figure supplement 3A and B*), it even aggravated their ampicillin sensitivity (*Figure 4—figure supplement 3C*). Under these conditions, wild-type cells also became more sensitive to the antibiotic, indicating that ampicillin efficacy is influenced by the metabolic state of the cells or the composition of the culture medium. Together, the results obtained demonstrate that PG recycling is important for proper growth and β-lactam resistance in *C. crescentus*.

Notably, previous work has shown that inactivation of the divisome-associated soluble lytic transglycosylase SdpA also leads to a severe decrease in ampicillin resistance in otherwise wild-type cells (*Zielińska et al., 2017*). This observation suggested that SdpA could represent the main lytic transglycosylase in *C. crescentus*, producing the largest share of the anhydro-muropeptides that are fed into the PG recycling pathway. Consistent with this notion, cells lacking SdpA were highly sensitive to even low concentrations of ampicillin (*Figure 4—figure supplement 4A*). Surprisingly, however, their resistance defect was more pronounced than that of any of the PG recycling-deficient mutants investigated (compare *Figure 4C*). Moreover, in accordance with previous work, Δ*sdpA* cells did not show any signs of cell filamentation (*Zielińska et al., 2017*), indicating that their reduced ampicillin resistance may not be primarily due to a decrease in the supply of anhydro-muropeptides available for PG recycling (*Figure 4—figure supplement 4B, C*). Metabolomics analysis indeed showed that the absence of SdpA barely affected the levels of cytoplasmic anhMur*N*Ac-peptide species accumulating in the Δ*amiR* background (*Figure 4—figure supplement 4D*). This finding supports the idea that SdpA is not the main producer of anhydro-muropeptides and likely affects β-lactam resistance through a different pathway, e.g., by preventing the toxic accumulation of non-crosslinked PG strands that form excessively when the TPase domain of bifunctional PBPs is inactivated by β-lactam binding (*Cho et al., 2014*; *Weaver et al., 2022*).

## The recycling of GlcNAc involves two partially redundant enzymatic systems

After removal of the peptide and cleavage of the GlcNAc–anhMur*N*Ac moiety, the two sugars are recycled in different ways. In *E. coli*, GlcNAc is transformed to GlcNAc-6-P by NagK, which is then reintroduced into de novo PG biosynthesis by conversion into GlcN-6-P, catalyzed by the deacetylase NagA (*Figure 1*). Bioinformatic analysis revealed that this branch of the PG recycling pathway is conserved in *C. crescentus*. Interestingly, however, this species contains two NagA homologs (75% identity), which we have named NagA1 and NagA2 (*Table 1*). To determine the physiological relevance of GlcNAc recycling, we generated *C. crescentus* Δ*nagK*, Δ*nagA1*, and Δ*nagA2* mutants and analyzed their cell morphologies and resistance to ampicillin. None of the strains showed a significant morphological (*Figure 5A and B*) or resistance (*Figure 5C*) phenotype, again suggesting that the supply of sugars is not a major constraining factor in PG biosynthesis. Since it was conceivable that NagA1 and NagA2 had redundant functions, we also aimed to generate a double mutant lacking both *nagA* homologs to fully interrupt the second reaction of the GlcNAc recycling branch. Surprisingly, this was not possible, even though blocking the first step by deletion of *nagK* did not cause any obvious fitness defect, suggesting that the accumulation of GlcNAc-6-P in the complete absence of NagA activity (*Plumbridge, 2009*) may be toxic to the cells, possibly by allosterically activating the GlcN-6-P deaminase NagB (*Álvarez-Añorve et al., 2016*) and thus preventing de novo PG biosynthesis.

In *E. coli*, the expression of *nagA* is controlled by GlcNAc-6-P, which interacts with the transcription factor NagC, thereby converting it from a repressor to an activator (*Plumbridge, 1991*). The system can also be activated by periplasmic GlcNAc, because *E. coli* possesses a phosphotransferase system (PTS) permease (NagE) that takes up and concomitantly phosphorylates this sugar, releasing it in the cytoplasm as GlcNAc-6-P (*Mukhija and Erni, 1996*; *Plumbridge, 2009*). Since *C. crescentus* possesses two *nagE* homologs (*Table 1*), we decided to analyze its response to externally added GlcNAc to better understand the regulation of NagA1 and NagA2 in this species. Comparing the total proteomes of wild-type cells grown in minimal (M2G) medium with or without GlcNAc, we found that the presence of the sugar strongly stimulated the expression of two distinct but similar gene clusters,

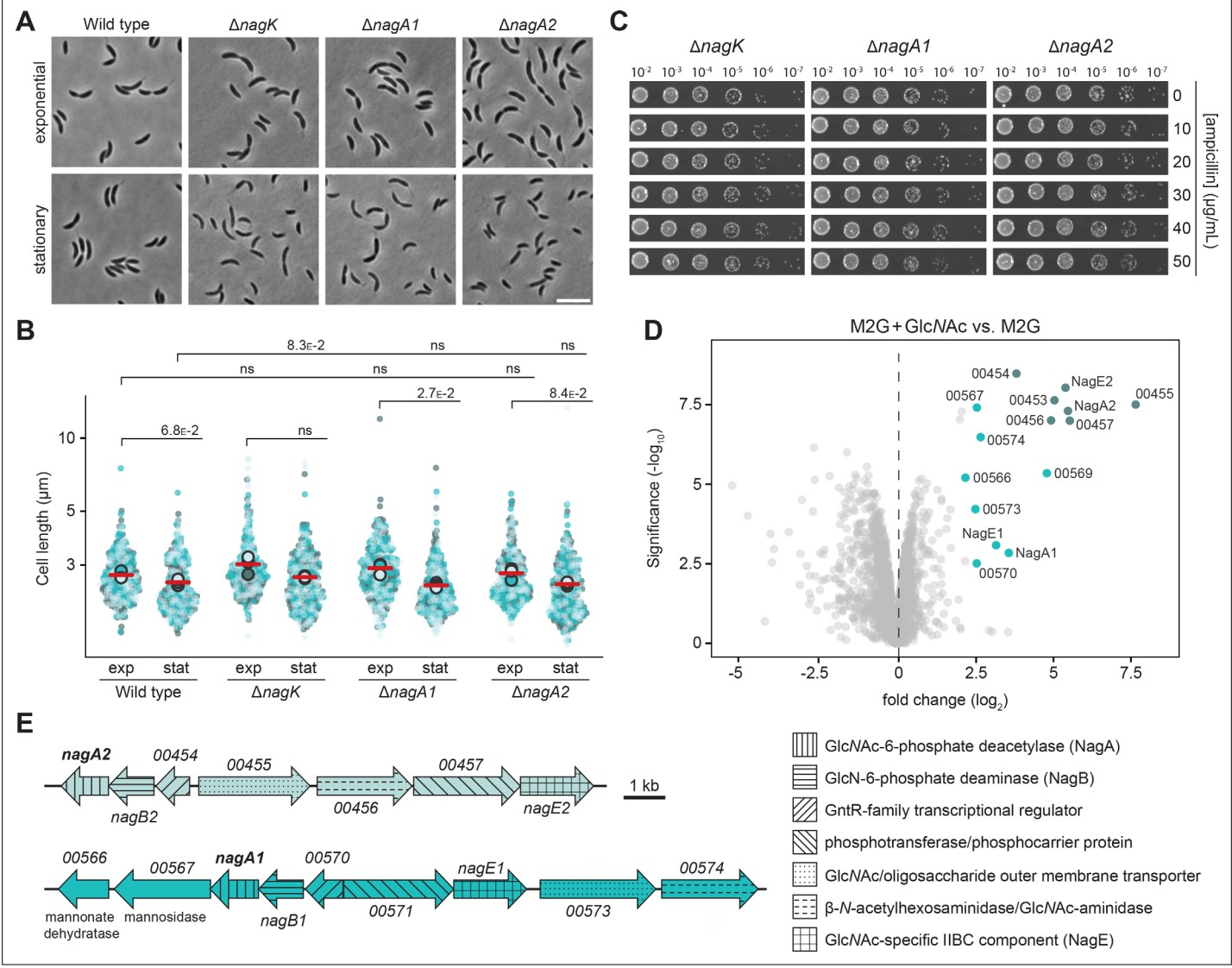

**Figure 5.** The *N*-acetylglucosamine (GlcNAc) recycling pathway is dispensable for proper cell division and β-lactam resistance. (**A**) Phase contrast images of *C. crescentus* wild-type, Δ*nagK* (PR255), Δ*nagA1* (PR256), and Δ*nagA2* (PR257) cells harvested in the exponential and stationary growth phase. Scale bar: 5 μm. (**B**) Superplots showing the distribution of cell lengths in populations of *C. crescentus* wild-type, Δ*nagK* (PR255), Δ*nagA1* (PR256), and Δ*nagA2* (PR257) cells in the exponential (exp) and stationary (stat) growth phase. Data (n=100 cells per replicate) are presented as described for *Figure 4B*. The statistical significance (*p*-value) of differences between conditions was assessed using a two-sided Welch's *t*-test. ns indicates *p*-values >0.1. (**C**) Serial dilution spot assays investigating the growth of *C. crescentus* Δ*nagK* (PR255), Δ*nagA1* (PR256), and Δ*nagA2* (PR257) cells on agar plates containing different concentrations of ampicillin. The cells were spotted on the same plates as those depicted in *Figure 4C*. See *Source data 1* for all replicates (n=3 independent experiments). (**D**) Volcano plot illustrating differential protein abundance in *C. crescentus* wild-type cells grown in M2G minimal medium with GlcNAc compared to plain M2G medium. Gray dots represent proteins identified by mass spectrometry. The x-axis indicates the log2 of the average difference in the peptide counts for the two different conditions (n=3 independent replicates). The y-axis shows the -log10 of the corresponding *p*-value. Proteins encoded in the two *nag* gene clusters are highlighted in color and annotated. The colors correspond to those used in panel E. (**E**) Schematic representation of the two *nag* gene clusters. The predicted functions of the gene products are specified in the legend on the right.

containing *nagA1* and *nagA2*, respectively (*Figure 5D and E*). One of these clusters (CCNA_00452-CCNA_00458) has previously been implicated in the uptake and metabolism of GlcNAc in *C. crescentus*, based on the fact that it includes a gene for an outer-membrane TonB-dependent receptor that transports GlcNAc and is essential for its utilization (*Eisenbeis et al., 2008*). The other genes in the cluster encode a NagE-like PTS permease, its cognate PTS phosphotransferase, NagA2 and a NagB homolog (*Figure 5E*), thus providing the complete set of proteins required to import GlcNAc

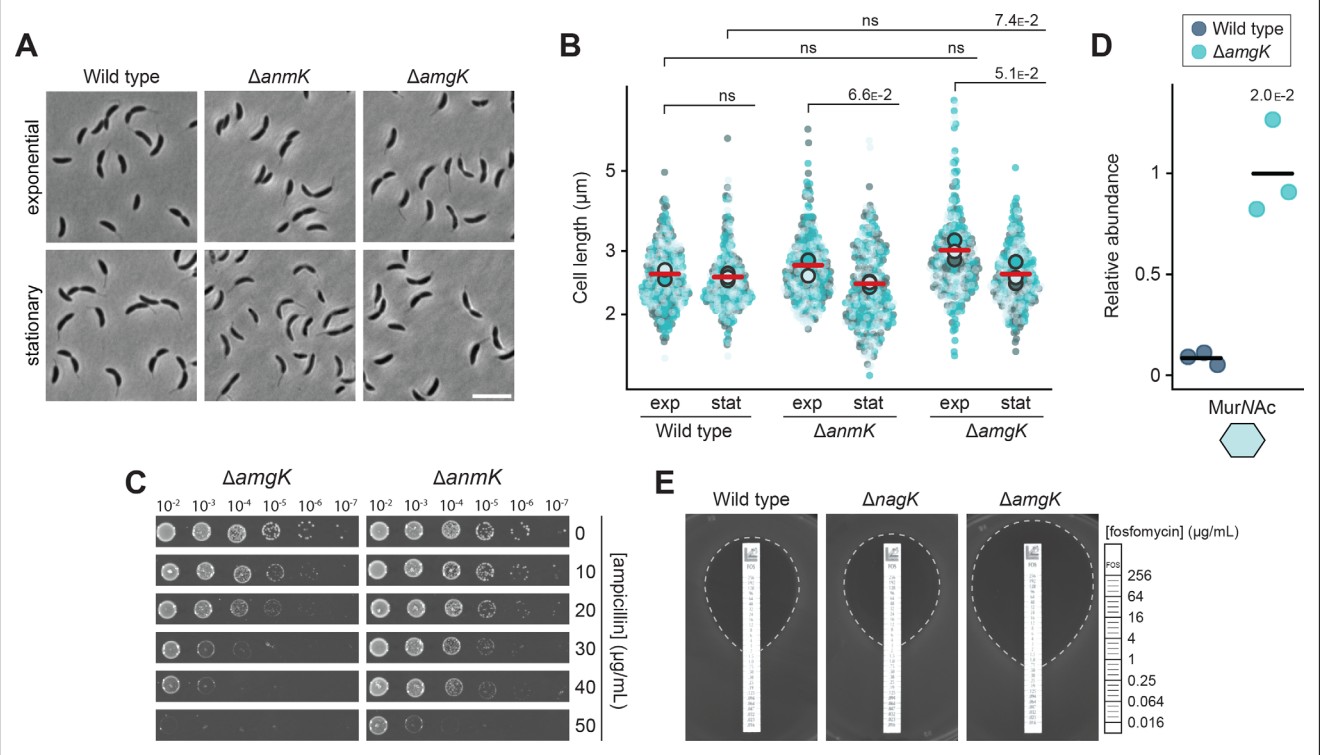

**Figure 6.** *C. crescentus* recycles anhydro-*N*-acetylmuramic acid (anhMur*N*Ac) through the MurU pathway. (**A**) Phase contrast images of *C. crescentus* wild-type, Δ*anmK* (PR252), and Δ*amgK* (PR262) cells, harvested in the exponential and stationary growth phase. Scale bar: 5 μm. (**B**) Superplots showing the distribution of cell lengths in populations of *C. crescentus* wild-type, Δ*anmK* (PR252), and Δ*amgK* (PR262) cells in the exponential (exp) and stationary (stat) growth phase. Data (n=100 cells per replicate) are presented as described for *Figure 4B*. The statistical significance (*p*-value) of differences between conditions was assessed using a two-sided Welch's *t*-test. ns indicates *p*-values >0.1. (**C**) Serial dilution spot assay investigating the growth of *C. crescentus* Δ*anmK* (PR252) and Δ*amgK* (PR262) on plates containing different concentrations of ampicillin. The cells were spotted on the same plates as those depicted in *Figure 4C*. See *Source data 1* for all replicates (n=3 independent experiments). (**D**) Levels of N-acetylmuramic acid (Mur*N*Ac) in the cytoplasm of *C. crescentus* wild-type and Δ*amgK* (PR262) cells, measured by metabolomics analysis through quantification of the corresponding mass spectrometric peak areas. The mass spectrometric peak areas were normalized against the mean obtained for the Δ*amgK* mutant. The statistical significance of differences between the two strains was determined using a two-sided Welch's *t*-test. (**E**) Analysis of the growth of *C. crescentus* wild-type, Δ*nagK* (PR255), and Δ*amgK* (PR262) cells on agar containing a fosfomycin gradient. The fosfomycin concentrations on the MIC test strip are indicated in the legend on the right.

from the periplasm and then funnel it either into PG biosynthesis or central carbon metabolism (see *Figure 1*). Finally, the cluster contains genes for a β-*N*-acetylhexosaminidase, catalyzing the release of Glc*N*Ac from chitin oligomers, and a transcriptional regulator of unknown function, which does not have an obvious homolog in *E. coli*. The second cluster (CCNA_00566-CCNA_00574) includes homologs of all the above-mentioned genes and additionally features genes for a predicted mannosidase and a predicted mannonate dehydratase, suggesting that it could be additionally involved in the metabolism of mannose derivatives. Collectively, the results of the mutational and proteomics analyses suggest that NagA1 and NagA2 are functionally redundant enzymes whose synthesis is activated by their substrate Glc*N*Ac-6-P.

## anhMur*N*Ac is recycled through the MurU pathway

Bioinformatic analysis suggested that *C. crescentus* contains all enzymes of the *P. aeruginosa* MurU pathway, which recycles anhMur*N*Ac independently of the Glc*N*Ac recycling branch, converting it directly back to UDP-Mur*N*Ac (*Borisova et al., 2017*; *Fumeaux and Bernhardt, 2017*; *Gisin et al., 2013*; *Figure 1* and *Table 1*). To verify the functionality of this predicted pathway and clarify its impact on cell physiology, we introduced deletions in the *anmK* or *amgK* gene, thereby blocking essential steps in the reaction sequence. Microscopic analysis revealed that the resulting single mutants did not have any significant morphological defects (*Figure 6A and B*). Importantly, however, both strains

showed a moderate increase in their susceptibility to ampicillin (*Figure 6C*). This finding indicates that the MurU pathway is active and relevant for β-lactam resistance in *C. crescentus*, thus playing a more important role than the GlcNAc recycling branch (compare *Figure 5A–C*). To determine whether this pathway was the only means for *C. crescentus* to metabolize anhMurNAc, we tested whether its disruption would lead to an accumulation of recycling intermediates. Metabolomics analysis indeed revealed a considerable increase in the cytoplasmic level of MurNAc in cells lacking the MurNAc kinase AmgK, suggesting the absence of alternative recycling pathways for the anhMurNAc moiety (*Figure 6D*). Notably, previous work has shown that the formation of UDP-MurNAc from recycled anhMurNAc by the MurU pathway reduces the sensitivity of *P. aeruginosa* to the antibiotic fosfomycin, which inhibits MurA and thus hinders the de novo synthesis of this central PG precursor (*Borisova et al., 2014*). We reasoned that this effect should also be observable in *C. crescentus* if the MurU pathway was active in this species. Antibiotic resistance assays revealed that the ΔamgK mutant indeed showed higher sensitivity to fosfomycin than the wild-type strain or the ΔnagK mutant, which is defective in GlcNAc recycling (*Figure 6E*). Taken together, these findings strongly suggest that anhMurNAc is recycled via the MurU pathway in *C. crescentus*, thereby enhancing the supply of precursors for peptidoglycan biosynthesis. This process may be more significant than GlcNAc recycling (compare *Figure 5A–C*)

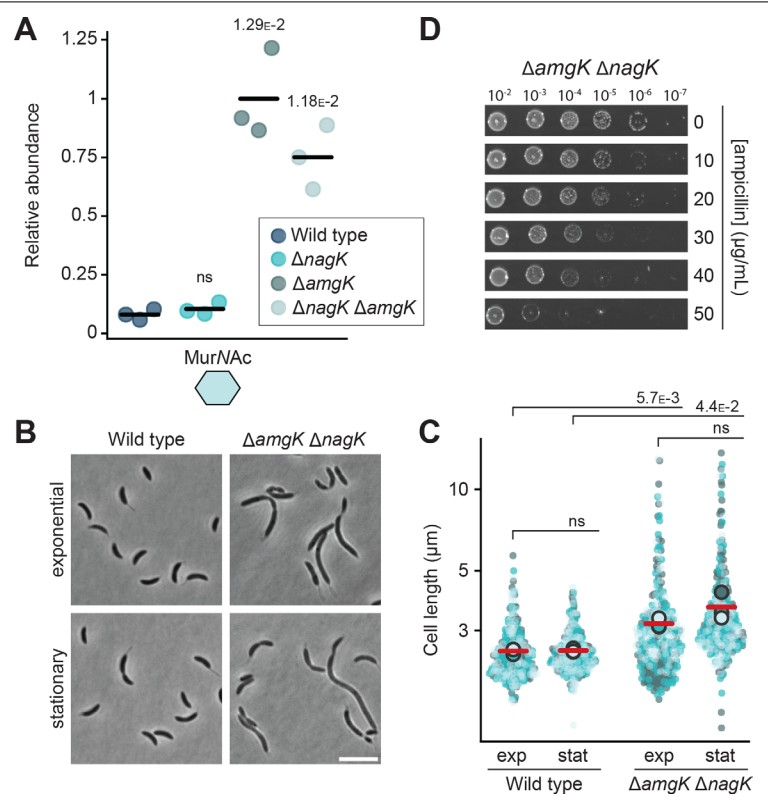

**Figure 7.** Recycling of the GlcNAc–anhMurNAc disaccharide moiety plays only a minor role in cell division and β-lactam resistance. (**A**) Levels of MurNAc in the cytoplasm of *C. crescentus* wild-type, ΔnagK (PR255), ΔamgK (PR262), and ΔnagK ΔamgK (PR261) cells, as determined by metabolomics analysis. Data were normalized to the mean MurNAc levels of ΔamgK cells. The statistical significance of differences between strains was determined using a two-sided Welch's *t*-test. ns indicates *p*-values <0.1. (**B**) Phase contrast images of *C. crescentus* wild-type and ΔamgK ΔnagK (PR261) cells, harvested in the exponential and stationary growth phase. Scale bar: 5 µm. (**C**) Superplots showing the distribution of cell lengths in populations of *C. crescentus* wild-type and ΔamgK ΔnagK (PR261) cells in the exponential (exp) and stationary (stat) growth phase. Data (n=100 cells per replicate) are presented as described for *Figure 4B*. The statistical significance (*p*-value) of differences between conditions was assessed using a two-sided Welch's *t*-test. ns indicates *p*-values >0.1. (**D**) Serial dilution spot assays investigating the growth of *C. crescentus* ΔamgK ΔnagK (PR261) cells on agar plates containing different concentrations of ampicillin. The cells were spotted on the same plates as those depicted in *Figure 4C*. See *Source data 1* for all replicates (n=3 independent experiments).

because a reduced production of GlcN-6-P may be compensated by an increased production of this intermediate from Fru-6-P, supplied by central carbon metabolism, or by import of GlcNAc from the growth medium.

As the recycling of GlcNAc and anhMurNAc is mediated by two distinct metabolic routes, cells blocked in one of these routes can still recycle half of the sugars generated by anhydro-muropeptide processing, potentially explaining the relatively mild phenotypes observed for the different single mutants. To clarify the effect caused by inhibiting both sugar recycling pathways, we generated a ΔnagK ΔamgK double mutant. Metabolomic analysis confirmed that this strain showed highly elevated MurNAc levels, comparable to those in the ΔamgK single mutant, while this recycling intermediate was barely detectable in ΔnagK or wild-type cells (*Figure 7A*). The double mutant showed a slightly higher tendency to form filaments than the wild type (*Figure 7B and C*). However, its susceptibility to ampicillin was not reduced beyond the level seen for the ΔamgK single mutant (*Figure 7D*). This observation may be attributed to the presence of GlcNAc in the rich medium used to grow the cells, which can be taken up and concomitantly phosphorylated by the NagE1 and NagE2 permease systems, thereby compensating for the loss of the GlcNAc kinase activity normally provided by NagK (see *Figure 1*).

## Defects in PG recycling do not change the levels or activity of the β-lactamase BlaA

In gammaproteobacteria, such as *P. aeruginosa* and *C. freundii*, β-lactam resistance is established by increased synthesis of the β-lactamase AmpC in the presence of excess anhydro-muropentapeptide, mediated by the transcriptional regulator AmpR (*Dietz et al., 1997*; *Dik et al., 2018*; *Jacobs et al., 1997*). Previous work has suggested that the *C. crescentus* β-lactamase BlaA is produced constitutively (*Docquier et al., 2002*; *Modi et al., 2025*), raising the question of how its protective effect could be dependent on PG recycling. To address this conundrum, we aimed to reinvestigate the regulation of BlaA activity at the transcriptional, protein accumulation, and activity level.

Inspecting the genomic contexts of PG recycling genes, we found that *amiR* is located in a three-gene operon that also includes genes for a predicted drug/metabolite transporter (named TraX) and a PaiB-family transcriptional regulator (named RegX), suggesting that these proteins could be involved in PG recycling and/or the regulation of β-lactam resistance (*Figure 8—figure supplement 1A*). However, in line with the dominant role of the permease AmpG in anhydro-muropeptide import (see *Figure 3*), the absence of TraX did not markedly affect the accumulation of anhMurNAc-peptides observed in the ΔamiR background. Similarly, whole-cell proteome analysis showed that the regulon of RegX did not include *blaA* or any genes related to PG recycling. Apart from this, neither ΔtraX nor ΔregX cells showed any obvious defects in cell division or ampicillin resistance, indicating that these two genes are not involved in PG-related processes (*Figure 8—figure supplement 1B–F*).

Interestingly, the *blaA* gene is located in a five-gene operon, which additionally encodes a predicted transcriptional regulator and three proteins involved in methionine biosynthesis (*Figure 8A*). To test whether the expression of this operon changes in response to β-lactam stress or disruption of the PG recycling pathway, we fused its upstream region (400 bp) to a β-galactosidase reporter gene (*lacZ*) and analyzed it for promoter activity in the ΔampG, ΔamiR, and ΔnagZ backgrounds both in the absence or in the presence of ampicillin. The results showed that *lacZ* expression was constant in all conditions tested (*Figure 8B*), indicating that the operon as a whole was not subject to transcriptional regulation. However, it was possible that the transcriptional regulator controlled *blaA* expression by acting on an internal promoter. To test this hypothesis, we compared the whole-cell proteomes of the wild-type strain and a mutant lacking the regulatory gene (CCNA_02225). Surprisingly, in the mutant strain, the methionine biosynthetic proteins encoded by the genes flanking *blaA* were significantly upregulated, whereas the levels of BlaA remained essentially unchanged (*Figure 8—figure supplement 2*). This finding may suggest that *blaA* is expressed from a strong and constitutive internal promoter, so that moderate changes in the transcription of the whole operon do not markedly affect the overall levels of the *blaA* transcripts (*Plumbridge, 1996*). Finally, we tested whether the activity of BlaA could be regulated post-transcriptionally at the level of enzyme activity. To this end, we incubated cells with the chromogenic β-lactam nitrocefin (*O'Callaghan et al., 1972*) and used its property of developing red color upon cleavage to measure β-lactamase activity in vivo. We found the ΔampG, ΔamiR, and ΔnagZ mutants showed behaviors similar to that of the wild-type strain,

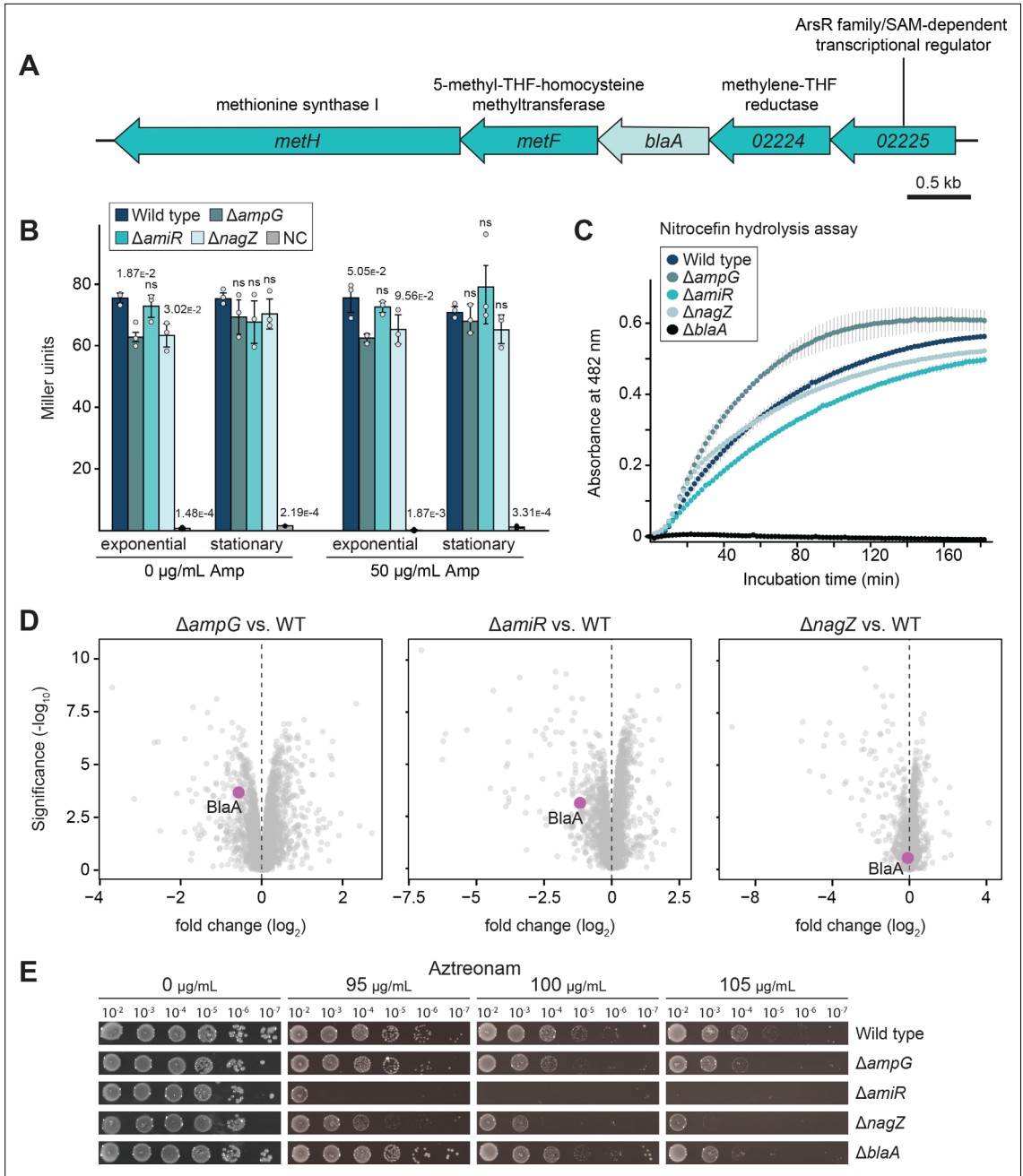

**Figure 8.** BlaA function is not regulated at the levels of transcription, protein accumulation, or enzymatic activity. (**A**) Organization of the putative five-gene operon containing the metallo-β-lactamase (*blaA)* gene. The annotations or ORF numbers of the open reading frames and the predicted functions of their gene products are indicated. (**B**) Activity of the promoter likely driving the expression of the *blaA*-containing operon in different strain backgrounds. The *C. crescentus* wild-type, Δ*ampG* (PR207), Δ*amiR* (PR033), and Δ*nagZ* (PR188) strains were transformed with a plasmid carrying a 400 bp fragment of the CCNA_02225 upstream region fused to a β-galactosidase reporter gene and assayed for reporter activity. Strains transformed with the empty plasmid served as negative controls (NC). The underlying data are shown as gray dots. The statistical significance (*p*-value) of differences between strains was calculated using a two-sided Welch's *t*-test. ns indicates *p*-values >0.1. (**C**) β-lactamase activity of *C. crescentus* wild-type, Δ*ampG* (PR207), Δ*amiR* (PR033), and Δ*nagZ* (PR188) cells, as determined by monitoring the hydrolysis of the chromogenic β-lactam nitrocefin in permeabilized cells over time. (**D**) Volcano plots showing differential protein abundance in *C. crescentus* Δ*ampG* (PR207), Δ*amiR* (PR033), and Δ*nagZ* (PR188) cells compared to wild-type cells. Gray dots represent proteins identified by mass spectrometry. The x-axis indicates the log2 of the average difference in the peptide counts for the two different conditions (n=4 independent replicates). The y-axis shows the -log10 of the corresponding *p*-value. The BlaA protein is highlighted in magenta. (**E**) Serial dilution spot assays investigating the growth of *C. crescentus* wild-type, Δ*ampG* (PR207), Δ*amiR* (PR033), Δ*nagZ* (PR188), and Δ*blaA* cells (CS606) on agar plates containing different concentrations of aztreonam. See ***Source data 1*** for all replicates (n=3 independent experiments).

*Figure 8 continued on next page*

*Figure 8 continued*

The online version of this article includes the following figure supplement(s) for figure 8:

**Figure supplement 1.** The two genes forming an operon with *amiR* are dispensable for peptidoglycan recycling or β-lactam resistance.

**Figure supplement 2.** Changes in protein accumulation upon deletion of CCNA_02225.

while the Δ*blaA* mutant lacked activity, validating the assay (*Figure 8C*). Consistent with this finding, whole-cell proteomic analyses showed that cells impaired in PG recycling accumulated BlaA at levels comparable to those in the wild-type strain (*Figure 8D*). Collectively, these results demonstrate that the severe reduction in β-lactam resistance observed in PG recycling-deficient strains is not due to changes in BlaA accumulation or activity. This conclusion is supported by the results of growth assays showing that defects in the PG recycling pathway also reduce the resistance of cells to aztreonam, a β-lactam antibiotic that is not cleaved by metallo-β-lactamases, such as BlaA (*Bush et al., 1982*; *Figure 8E*). Thus, the ampicillin sensitivity phenotypes observed are likely due to changes in the susceptibility and/or activity of target proteins rather than impaired degradation of this antibiotic in the absence of PG recycling.

## PG recycling defects sensitize the PG biosynthetic machinery to the inhibitory effect of ampicillin

To shed more light on the connection between PG recycling and β-lactam resistance, we aimed to investigate the growth defects induced by the exposure of cells to ampicillin in more detail. To this end, we cultivated wild-type and Δ*amiR* cells in liquid medium containing different concentrations of the antibiotic and analyzed their morphology by light microscopy. Wild-type cells tolerated substantial concentrations of ampicillin without noticeable changes in cell shape (*Figure 9A and B*). However, at 500 µg/mL, they developed a pronounced filamentous phenotype, while exposure to 5000 µg/mL led to growth arrest and cell lysis. For the Δ*amiR* mutant, similar phenotypes were observed, but cell filamentation and lysis already occurred at the lowest ampicillin concentration tested and were consistently more pronounced than for the wild type across all conditions (*Figure 9A and B*). To better characterize the development of the observed morphological defects, we monitored the growth behavior of the two strains following a shift from plain rich medium to medium supplemented with a low concentration of ampicillin (20 µg/mL). Whereas the wild-type strain grew normally in both conditions (*Figure 9C* and *Figure 9—video 1*), Δ*amiR* cells started to become filamentous and swollen shortly after exposure to the antibiotic and were unable to divide (*Figure 9C* and *Figure 9—video 2*). Together, these results indicate that, in *C. crescentus*, ampicillin preferentially targets cell division, although it also affects lateral growth when applied at elevated concentrations. Moreover, they suggest that PG recycling defects enhance the efficacy of ampicillin by rendering the PG biosynthetic machinery more sensitive to its inhibitory effect.

To shed more light on the inhibitory activity of ampicillin, we aimed to identify its target proteins. For this purpose, we adapted an assay enabling the visualization of PBPs in SDS-gels after labeling with the fluorescent β-lactam bocillin-FL (*Zhao et al., 1999*). Using Δ*blaA* cells to prevent β-lactam degradation, this approach detected all known functional transpeptidases of *C. crescentus* except for PBP2 (*Strobel et al., 2014*; *Yakhnina and Gitai, 2013*) as well as the predicted carboxypeptidase CrbA (*Figure 9D*). However, when these proteins were incubated with ampicillin prior to the addition of bocillin-FL, all but one of them (PbpX) remained unlabeled, indicating that ampicillin had blocked their active-site serine, thereby preventing the subsequent reaction with bocillin-FL. Among the ampicillin targets identified were the essential, cell division-specific monofunctional transpeptidase PBP3 as well as the bifunctional PBPs Pbp1A, PbpY, and PbpC. The labeling of PBP3 and some bifunctional PBPs was also abolished by preincubation with cephalexin, a β-lactam that specifically targets PBP3 in *E. coli* (*Kocaoglu and Carlson, 2015*), whereas the PBP2-specific β-lactam mecillinam had only minor effects on the labeling pattern (*Figure 9D*). Given that Pbp1A, PbpY, and PbpC have been shown to be dispensable for cell division in *C. crescentus* (*Strobel et al., 2014*; *Yakhnina and Gitai, 2013*), these results indicate that the cell filamentation phenotype induced by ampicillin is primarily due to inhibition of PBP3 activity.

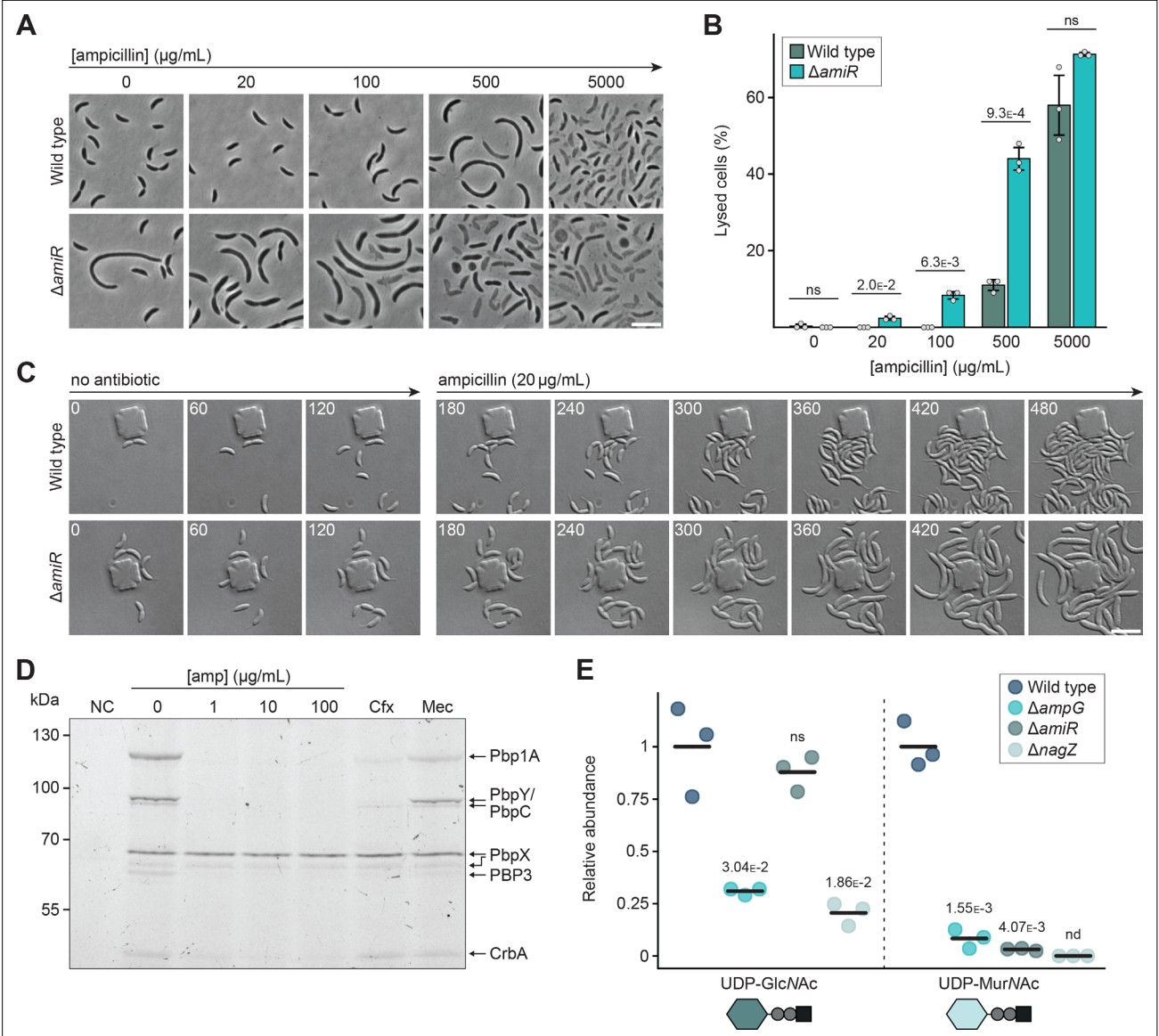

**Figure 9.** Peptidoglycan (PG) recycling defects increase the sensitivity of the septal PG synthetic machinery to ampicillin and reduce PG precursor biosynthesis. (**A**) Phase contrast images of *C. crescentus* wild-type and Δ*amiR* (PR033) cells, incubated for 3 hr in the presence of different concentrations of ampicillin. Scale bar: 5 µm. (**B**) Quantification of the fraction of lysed cells in cultures of *C. crescentus* wild-type and Δ*amiR* (PR033) cells, incubated for 3 hr in the presence of different concentrations of ampicillin. The bars represent the mean (± SD) of three independent replicates (n=100 cells per condition). The underlying data are shown as gray dots. The statistical significance (*p*-value) of differences between strains was calculated using a two-sided Welch's *t*-test. ns indicates *p*-values >0.1. (**C**) Time-lapse microscopy analysis showing *C. crescentus* wild-type and Δ*amiR* (PR033) cells in a microfluidic flow cell before and after ampicillin treatment. After 3 hr of growth in antibiotic-free medium, the cells were shifted to medium containing 20 µg/mL ampicillin. Images were taken at 5 min intervals for a total duration of 8 hr. Shown are representative frames of the time-lapse series. The full sequences are shown in *Figure 9—video 1* and *Figure 9—video 2*. (**D**) Image of an SDS-gel showing the labeling of penicillin-binding proteins (PBPs) with the fluorescent β-lactam bocillin-FL in the absence or presence of other β-lactams. Crude membrane fractions of *C. crescentus* Δ*blaA* (CS606) cells were pre-incubated with the indicated concentrations of ampicillin, 5 µg/mL cefalexin (Cfx), or 100 µg/mL mecillinam (Mec), treated with bocillin-FL, and then subjected to SDS gel electrophoresis to separate the labeled PBPs prior to fluorescence imaging. (**E**) Levels of UDP-GlcNAc and UDP-MurNAc in the cytoplasm of *C. crescentus* wild-type, Δ*ampG* (PR207), Δ*amiR* (PR033), and Δ*nagZ* (PR188) cells, measured by metabolomics analysis through quantification of the corresponding mass spectrometric peak areas. For each PG precursor, the mass spectrometric peak area was normalized against the mean obtained for the wild-type strain. The statistical significance of differences between the wild type and the mutant strains was determined using a two-sided Welch's *t*-test. ns indicates *p*-values >0.1; nd: not detectable.

The online version of this article includes the following video, source data, and figure supplement(s) for figure 9:

**Source data 1.** Annotated raw image of the SDS-gel shown in *Figure 9D*.

*Figure 9 continued on next page*

*Figure 9 continued*

**Source data 2.** Raw image of the SDS-gel shown in **Figure 9D**.

**Figure supplement 1.** Muropeptide composition of peptidoglycan (PG) sacculi from wild-type and Δ*amiR* cells.

**Figure supplement 2.** Ampicillin sensitivity of peptidoglycan recycling-defective mutants in the presence of *N*-acetylglucosamine (Glc*N*Ac).

**Figure 9—video 1.** Effect of ampicillin on the growth of *C. crescentus* wild-type cells.

https://elifesciences.org/articles/109465/figures#fig9video1

**Figure 9—video 2.** Effect of ampicillin on the growth of *C. crescentus* Δ*amiR* cells.

https://elifesciences.org/articles/109465/figures#fig9video2

## PG recycling defects may increase ampicillin sensitivity by limiting the supply of PG precursors

The results obtained suggest that impaired PG recycling does not affect BlaA activity and may rather enhance the inhibitory effect of ampicillin on PG biosynthesis. It was conceivable that the accumulation of unrecycled PG degradation products in the periplasm could impair the activity of certain PG biosynthetic enzymes, thereby disrupting PG homeostasis. However, sacculi isolated from wild-type and Δ*amiR* cells did not show any significant differences in their muropeptide composition (*Figure 9—figure supplement 1*), suggesting that a defect in PG recycling does not perturb the global balance of PG biosynthetic and autolytic processes. Notably, mutants blocked in the PG recycling pathway also exhibit a conditional cell division defect, even when not exposed to ampicillin. It was, therefore, possible that their higher β-lactam sensitivity was linked to a decrease in the overall efficiency of PG biosynthesis, caused by impaired replenishment of PG precursors from recycling products. Metabolomics analysis revealed that the Δ*ampG*, Δ*amiR*, and Δ*nagZ* mutants indeed showed significantly lower levels of UDP-sugars than the wild-type strain (*Figure 9E*). The absence of AmpG and NagZ led to a more than 50% decrease in the UDP-Glc*N*Ac pool, confirming that de novo synthesis alone is not sufficient to satisfy the cellular demand for this central metabolite. By contrast, UDP-Glc*N*Ac levels remained unchanged in Δ*amiR* cells, consistent with the observation that Glc*N*Ac can still be released from imported anhydro-muropeptides in the absence of amidase activity (see *Figures 2C and 3C*). Even stronger effects were observed on the UDP-Mur*N*Ac pool, which was drastically reduced in all three mutant strains (*Figure 9E*). Its depletion in the absence of PG recycling suggests that the formation of UDP-Mur*N*Ac constitutes a bottleneck in the de novo PG biosynthetic pathway, with cells relying heavily on anhMur*N*Ac recycling via the MurU pathway to maintain an adequate supply. Collectively, these results demonstrate that PG recycling is critical to provide sufficient activated sugars for PG precursor biosynthesis. However, although the Δ*ampG*, Δ*amiR*, and Δ*nagZ* mutants show a very similar decrease in cytoplasmic UDP-Mur*N*Ac levels, their morphological and resistance defects differ significantly, indicating that the reduced availability of UDP-sugars may not be the only reason for the phenotypes observed. Consistent with this notion, an increase in cytoplasmic Glc*N*Ac-P levels through supplementation of the growth medium with Glc*N*Ac was not sufficient to improve ampicillin tolerance in the three mutant strains (*Figure 9—figure supplement 2*). Likewise, the addition of the PG-specific sugar Mur*N*Ac did not markedly enhance growth in the presence of the antibiotic (*Figure 9—figure supplement 2B*), although it remains unclear whether *C. crescentus* can take up Mur*N*Ac from the environment.

It was conceivable that the limited supply of PG precursors in the absence of proper PG recycling causes cell filamentation and lysis by reducing the level of association between PBPs and their substrates, thereby slowing their catalytic activity. To test this idea, we investigated whether the cell division defect of the Δ*amiR* mutant could be ameliorated by artificially increasing the activity of the septal PG biosynthetic machinery. To this end, we employed a hyperactive (A246T) variant of the glycosyltransferase FtsW (FtsW*) that leads to increased activity of the septal FtsW-PBP3 PG biosynthetic complex (*Lariviere et al., 2019*; *Modell et al., 2014*), resulting in elevated cell constriction rates and shorter cell lengths, especially in stationary phase (*Figure 10A and B*). Interestingly, the introduction of the *ftsW** allele almost completely rescued the cell filamentation phenotype of the Δ*amiR* mutant in stationary phase, although it had only little effect in exponentially growing cells, possibly because PG precursors are more limiting under conditions of rapid growth. Thus, hyperactivation of the FtsW-PBP3 complex can, at least in part, reverse the cell division defect induced by defective PG recycling, suggesting that the cell filamentation phenotype observed in the mutant backgrounds

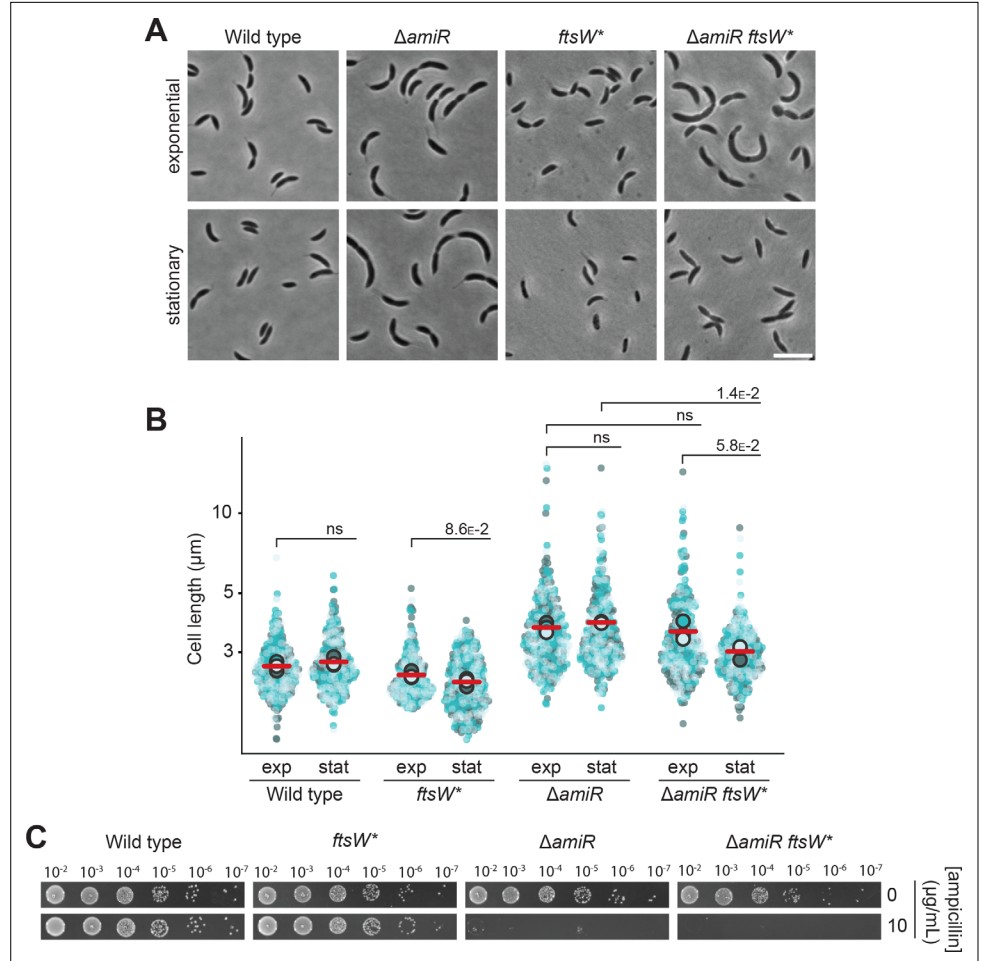

**Figure 10.** The cell filamentation phenotype of Δ*amiR* cells is rescued by hyper-activation of the septal FtsW-PBP3 peptidoglycan biosynthetic complex. (**A**) Phase contrast images of *C. crescentus* wild-type, Δ*amiR* (PR033), *ftsW::ftsW** (ML2103), and Δ*amiR ftsW::ftsW** (PR246) cells, harvested in the exponential and stationary growth phase. Scale bar: 5 μm. (**B**) Superplots showing the distribution of cell lengths in populations of the strains described in panel A in the exponential (exp) and stationary (stat) growth phase. Data (n=100 cells per replicate) are presented as described for *Figure 4B*. The statistical significance (*p*-value) of differences between conditions was assessed using a two-sided Welch's *t*-test. ns indicates *p*-values >0.1. (**C**) Serial dilution spot assay investigating the growth of *C. crescentus ftsW::ftsW** (ML2103), Δ*amiR* (PR033), and Δ*amiR ftsW::ftsW** (PR246) cells on agar plates containing no or 10 μg/mL ampicillin. See **Source data 1** for all replicates (n=3 independent experiments).

could indeed be due to a reduced activity of this complex. Notably, however, the presence of the *ftsW** allele did not relieve the ampicillin sensitivity of Δ*amiR* cells (*Figure 10C*), indicating that an increased activity of the septal PG biosynthetic complex does not reduce its ampicillin-mediated inactivation.

## Discussion

Bacterial PG biosynthesis has been studied intensively over the past decades. Importantly, apart from the proper spatiotemporal control of PG incorporation and cleavage during cell growth and cell division, the recycling of PG degradation products has emerged to play a major role in cellular fitness and survival. This process not only conserves valuable resources (*Gilmore and Cava, 2025*) but also plays an important role in infection by enabling bacteria to evade the human innate immune system (*Trindade and Chen, 2020*). Moreover, it was found to critically contribute to AmpR-mediated regulation of β-lactam resistance in *P. aeruginosa* and various enterobacterial pathogens (*Dik et al., 2018*). Although substantial advances have been made in elucidating the mechanisms and physiological roles

of PG recycling in gammaproteobacteria, these aspects remain incompletely characterized across other bacterial lineages (*Gilmore and Cava, 2025*). In this study, we demonstrate that the alphaproteobacterial model organism *C. crescentus* possesses a functional PG recycling pathway that includes components from both the *E. coli* and *P. aeruginosa* systems, extending the results of a recent study on this topic (*Modi et al., 2025*). We find that this pathway is required for maintaining an adequate supply of peptidoglycan precursors, complementing de novo synthesis. *C. crescentus* cells impaired in PG recycling show marked defects in cell division and cell wall integrity. Moreover, they exhibit a pronounced decrease in intrinsic β-lactam resistance, although the levels and activity of their major β-lactamase BlaA remain unchanged. These findings are in line with previous work in *A. tumefaciens* showing that a defect in PG recycling caused by blocking anhydro-muropeptide import impairs cell wall integrity and thereby reduces AmpC-mediated β-lactam resistance in a manner that extends beyond its effect on AmpC regulation (*Gilmore and Cava, 2022*). Blocking critical steps in the PG recycling pathway, therefore, emerges as a promising new strategy to combat β-lactam-resistant pathogens and thus mitigate the global antibiotic crisis.

Our analyses revealed that *C. crescentus* possesses almost all core PG recycling enzymes identified in gammaproteobacteria. Notable exceptions are the LD-carboxypeptidase LdcA (*Templin et al., 1999*), which trims sugar-linked and free tetrapeptides to tripeptides, as well as the murein peptide ligase Mpl, which re-attaches free peptide stems to UDP-MurNAc, thereby obviating the need for their de novo synthesis (*Mengin-Lecreulx et al., 1996*). In *E. coli*, the activity of LdcA is critical for cell envelope integrity, because its inactivation leads to the accumulation of free tetrapeptides, which are then erroneously ligated with UDP-MurNAc, giving rise to non-crosslinkable tetrapeptide-containing PG precursors, whose incorporation into peptidoglycan severely destabilizes the cell wall structure (*Templin et al., 1999*). In some gammaproteobacteria, the function of LdcA is taken over by other, unrelated peptidases (*Dai et al., 2021*; *Hernández et al., 2020*; *Simpson et al., 2021*), opening the possibility that *C. crescentus* possesses a thus-far unknown peptide-processing LD-carboxypeptidase. However, metabolomic analysis showed that Δ*amiR* cells accumulated high concentrations of anhMurNAc-tetrapeptide in their cytoplasm (*Figure 3B*), making the existence of such an enzyme unlikely. In line with the absence of Mpl, *C. crescentus* may thus not recycle tripeptides but rather degrade the peptide stems released by AmiR into individual amino acids, which are subsequently reused for de novo PG biosynthesis. A similar situation may be seen for many other Gram-negative and Gram-positive species that lack both an LdcA and an Mpl homolog (*Gilmore and Cava, 2025*).

Apart from the removal of the peptide stem by AmiR, the cleavage of the GlcNAc–anhMurNAc moiety by NagZ is a critical step in the initial processing of imported anhydro-muropeptide species. The results of our enzyme assays (*Figure 2*) and metabolomic analyses (*Figure 3*) show that these two reactions do not follow a defined order but occur independently of each other. Once released, the two amino sugars follow distinct metabolic routes. GlcNAc is phosphorylated to GlcNAc-6-P and then deacetylated by NagA, yielding GlcN-6-P, which is then either used as the starting point of de novo PG biosynthesis or, alternatively, fed into central carbon metabolism through deamination to Fru-6-P by NagB (*Figure 1*). This branch point is critical since GlcNAc is not only produced during peptidoglycan degradation but also acquired from the environment as a readily available carbon source, primarily derived from the breakdown of chitin (*Beier and Bertilsson, 2013*). In the presence of external GlcNAc, two related gene clusters are highly upregulated, each of them, including genes for an outer-membrane TonB-dependent receptor, a PTS permease together with its cognate phosphorylation system as well as a homolog of the deacetylase NagA and the deaminase NagB (*Figure 5D and E*). Their regulation is likely mediated by GntR-type transcriptional regulators encoded in the same clusters, which may sense the accumulation of GlcNAc-6-P as the first common cytoplasmic intermediate of the PG recycling and GlcNAc uptake pathways, thereby adapting the flux through the GlcNAc recycling/utilization pathway to substrate availability. While one of the TonB-dependent receptors (CCNA_00455) was shown to be specific for GlcNAc (*Eisenbeis et al., 2008*), the substrate range of the other transport systems still remains to be determined. Overall, the GlcNAc recycling/utilization system of *C. crescentus* shows parallels to that of *E. coli*, which also involves two different PTS permeases, NagE and ManXYZ, for GlcNAc transport (*Plumbridge and Vimr, 1999*). However, *E. coli* lacks a GlcNAc-specific TonB-dependent receptor, shows a different organization and regulation of the genes involved and only possesses a single set of NagA and NagB proteins. Even more pronounced differences are observed for the recycling of anhMurNAc, which in both species initiates

with the formation of anhMur*N*Ac-6-P. In *E. coli*, this intermediate is converted to Glc*N*Ac-6-P by the etherase MurQ and thus funneled into the Glc*N*Ac recycling pathway (*Jaeger et al., 2005*; *Uehara et al., 2006*). *C. cresentus*, by contrast, lacks this enzyme and instead uses the alternative MurU pathway, a shortcut that transforms Mur*N*Ac-6-P to UDP-Mur*N*Ac and thus feeds it back directly into PG biosynthesis (*Figures 6 and 7*). This pathway has so far only been observed in a few other species, such as the gammaproteobacteria *P. aeruginosa* (*Borisova et al., 2017*; *Fumeaux and Bernhardt, 2017*; *Gisin et al., 2013*) and *Legionella pneumophila* (*Ratna et al., 2025*) as well as *Tannerella forsythia* (*Hottmann et al., 2021*), a member of the Bacteroidia. Our results, combined with those of a previous study (*Modi et al., 2025*), show that it is more widespread than previously anticipated and also conserved in the alphaproteobacterial lineage.

In previously studied model species, PG recycling was found to be dispensable under laboratory conditions and defects in this process did not lead to any obvious morphological defects (*Gilmore and Cava, 2025*; *Park and Uehara, 2008*), except for the lack of LdcA, which causes cell rounding and lysis in stationary phase in *E. coli* (*Templin et al., 1999*). In *C. crescentus*, it was also possible to introduce single deletions in all PG recycling-related genes, although we were not able to create a strain lacking both *nagA* homologs simultaneously, suggesting that the accumulation of Glc*N*Ac-6-P may be toxic to the cells. Notably, however, and contrary to a previous report (*Modi et al., 2025*), most mutants displayed cell division defects and cell filamentation, especially when transitioning to stationary phase. These phenotypes were particularly pronounced for Δ*ampG* and Δ*amiR* cells, which are impaired in anhydro-muropeptide uptake or peptide stem cleavage, respectively. By contrast, single disruptions in the two sugar recycling branches barely affected cell length, suggesting that the observed filamentation phenotype is linked to impaired peptide recycling. Interestingly, while completely abolishing sugar recycling through inactivation of NagZ did not cause any obvious morphological defects, blocking both the Glc*N*Ac and anhMur*N*Ac recycling branches led to cell filamentation. This conundrum may be explained by the accumulation of metabolic intermediates in the double mutant that may cause misregulation of PG biosynthesis by affecting gene expression or enzyme activities. The precise link between PG recycling and cell division remains to be clarified. However, we observed that blocking the initial steps of PG recycling leads to a severe reduction in the levels of PG precursors, such as UDP-Glc*N*Ac and UDP-Mur*N*Ac (*Figure 9E*). This effect may be even more pronounced for the peptide-containing downstream products of the PG biosynthetic pathway (which were not detectable with our methods), in particular in mutant backgrounds preventing the recycling of the peptide stem. Importantly, the cell division defects caused by impaired PG recycling were alleviated by hyperactivation of the FtsW-PBP3 complex (*Figure 10*). Together, these findings suggest that reduced PG precursor levels impair the association of divisome-associated GTases and PBPs with their substrates, leading to lower catalytic rates and a slow-down of cell constriction. This effect may be further aggravated in the presence of ampicillin, because β-lactam-mediated inhibition of transpeptidase activity can hyperactivate the GTase domain of bifunctional PBPs. As a consequence, they produce an abundance of non-crosslinked glycan strands that are rapidly degraded after synthesis by lytic transglycosylases, leading to further depletion of the lipid II pool (*Banzhaf et al., 2012*; *Cho et al., 2014*; *Kohlrausch and Höltje, 1991*). Moreover, apart from decreasing PG synthase activity, defects in PG recycling may also cause an accumulation of anhydro-muropeptides in the periplasm, which could potentially interfere with normal PG biosynthesis by occupying binding sites or promoting non-productive side-reactions. Interestingly, in contrast to cell division, longitudinal growth still proceeds efficiently in the absence of PG recycling, possibly because it requires less PG incorporation than cell constriction and the formation of the new cell poles.

Defects in PG recycling not only caused cell filamentation but also strongly reduced the intrinsic resistance of *C. crescentus* to ampicillin. This phenomenon was most pronounced for mutants with defects in the first PG recycling steps that completely abolished peptide and/or sugar recycling (*Figure 4*). A milder effect was observed for mutants blocked in the MurU pathway, whereas disrupting Glc*N*Ac recycling did not produce any obvious ampicillin sensitivity phenotype. Our results suggest that the formation of UDP-Mur*N*Ac as the first committed step in PG biosynthesis represents a metabolic bottleneck (*Figure 9E*), making anhMur*N*Ac recycling through the MurU shunt essential for maintaining adequate PG precursor levels. UDP-Glc*N*Ac, by contrast, which is used in the biosynthesis in various cell envelope polysaccharides, can be more readily replenished because its biosynthesis is supported not only by de novo synthesis and PG recycling but also by the uptake of

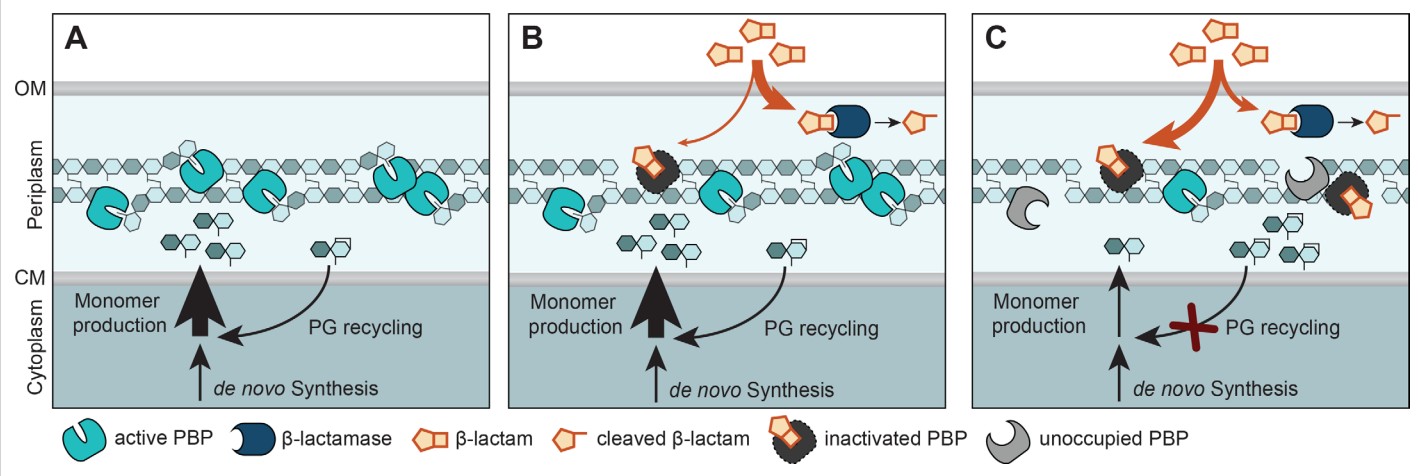

**Figure 11.** Model for the critical role of peptidoglycan (PG) recycling in *C. crescentus* growth and β-lactam resistance. (**A**) Penicillin-binding proteins (PBPs) mediate the incorporation of new cell wall material into the growing PG sacculus. The PG precursors required for this process are synthesized in the cytoplasm, using building blocks that are provided by both de novo synthesis and PG recycling. (**B**) If β-lactams enter the periplasm, they compete with PG precursors for binding to their PBP targets. Most antibiotic molecules are rapidly hydrolyzed by the metallo-β-lactamase (BlaA). However, a minor fraction manages to interact with PBPs before they are captured and degraded. The resulting inactivation of a small proportion of PBPs can be tolerated by cells, as long as they still contain sufficient active PBPs to maintain cell growth and division. (**C**) Upon disruption of PG recycling, the level of PG monomers strongly decreases, leaving the substrate binding sites of PBPs unoccupied for longer periods of time and thus increasing their accessibility to β-lactam molecules. As a result, a larger fraction of PBPs is inactivated, leading to decreased cell wall integrity and ultimately cell lysis. This effect may be aggravated by the reduced activity of the remaining active PBPs due to an insufficient supply of PG precursors and by the accumulation of anhydro-muropentapeptides in the periplasm, which could interfere with the activities of PG biosynthetic enzymes, thereby further decreasing the incorporation of new material into the existing sacculus.

external GlcNAc from the growth medium (*Eisenbeis et al., 2008*; *Figure 5D and E*). Notably, even high concentrations of external GlcNAc were not sufficient to restore ampicillin resistance in any of the PG recycling-deficient mutants tested (*Figure 9—figure supplement 2*). Contrary to a previous report (*Modi et al., 2025*), this finding supports the notion that their β-lactam sensitivity is not attributable to an inadequate supply of GlcNAc for PG precursor biosynthesis.

A role of PG recycling in β-lactam resistance has been previously reported for gammaproteobacteria such as *P. aeruginosa* and *C. freundii*. In these species, β-lactam exposure leads to an increase in the cytoplasmic level of anhMurNAc-pentapeptide, thereby stimulating its interaction with the transcriptional regulator AmpR and inducing expression of the *ampC* β-lactamase gene (*Dik et al., 2018*). However, recent work has indicated that different mechanisms may be at work in alphaproteobacteria (*Gilmore and Cava, 2022*; *Modi et al., 2025*). Our data show that, in *C. crescentus*, the accumulation and activity of the main β-lactamase BlaA remain unchanged in the presence of ampicillin and upon inhibition of PG recycling. The lower ampicillin resistance level of PG recycling-deficient mutants may therefore not be due to less efficient inactivation of ampicillin but rather an increased susceptibility of PBPs to this antibiotic. This effect could potentially be explained by competition of ampicillin and PG precursors for the substrate-binding sites of target PBPs (*Figure 11*). In wild-type cells, the abundant supply of precursor ensures the saturation of PBPs with their cognate substrate. As a consequence, most ampicillin molecules will be cleaved by BlaA before they manage to covalently modify their target proteins, leaving the vast majority of PBPs in a functional state. However, if defects in PG recycling limit the availability of PG precursors, the substrate binding sites of PBPs remain unoccupied for extended periods of time. As a consequence, they become more susceptible to interaction with residual ampicillin molecules, leading to an increase in the fraction of inactive enzymes. The resulting defects in cell division and cell wall integrity may be further aggravated by reduced activity of the remaining PBPs due to insufficient substrate availability and by the accumulation of periplasmic anhydro-muropeptides, which could block the substrate-binding sites of PG metabolic enzymes. Conversely, while interfering with cell division, the increased occupation of binding sites by excess anhydro-muropeptides may also hinder the interaction of ampicillin with its target PBPs – especially in the Δ*ampG* mutant – thereby reducing the susceptibility of cells to the antibiotic. In different PG

recycling-deficient mutants, these factors may be manifested to varying degrees, which could account for differences in the severity between the cell filamentation and ampicillin sensitivity phenotypes in many of the mutant strains.

Collectively, our results demonstrate that PG recycling is critical for proper growth, cell division and β-lactam resistance in *C. crescentus*, thus playing a more significant role in bacterial fitness and survival than previously appreciated. Moreover, they support the idea that bacterial β-lactam resistance is a complex trait, determined not only by specific resistance proteins but also by the supply of PG building blocks and the dynamics of PG biosynthesis. Given the widespread occurrence of PG recycling among bacteria, it will be important to determine the impact of this process on cell viability and antibiotic resistance in other species and to evaluate its inhibition as a potential new strategy to potentiate the efficacy of β-lactam antibiotics in antibacterial therapies.

# Methods

**Key resources table**

| Reagent type (species) or resource | Designation | Source or reference | Identifiers | Additional information |
|---|---|---|---|---|
| Gene (*Caulobacter crescentus*) | *amgK* | GenBank | ACL97114.1 | |
| Gene (*Caulobacter crescentus*) | *amiR* | GenBank | ACL96115.1 | |
| Gene (*Caulobacter crescentus*) | *ampG* | GenBank | ACL93603.1 | |
| Gene (*Caulobacter crescentus*) | *anmK* | GenBank | ACL95410.2 | |
| Gene (*Caulobacter crescentus*) | *CCNA_02225* | GenBank | ACL95690.1 | |
| Gene (*Caulobacter crescentus*) | *nagA1* | GenBank | ACL94033.1 | |
| Gene (*Caulobacter crescentus*) | *nagA2* | GenBank | ACL93919.1 | |
| Gene (*Caulobacter crescentus*) | *nagK* | GenBank | ACL97314.2 | |
| Gene (*Caulobacter crescentus*) | *nagZ* | GenBank | ACL95550.1 | |
| Gene (*Caulobacter crescentus*) | *regX* | GenBank | ACL96114.1 | |
| Gene (*Caulobacter crescentus*) | *traX* | GenBank | ACL96116.1 | |
| Strain, strain background (*Caulobacter crescentus*) | CB15N (aka NA1000) | *Evinger and Agabian, 1977* | ATCC 19089 | *C. crescentus* wild-type strain |
| Strain, strain background (*Caulobacter crescentus*) | AM399 | *Zielińska et al., 2017* | | CB15N Δ*sdpA* |
| Strain, strain background (*Caulobacter crescentus*) | CS606 | | | CB15N Δ*blaA* |
| Strain, strain background (*Caulobacter crescentus*) | ML2103 | *Modell et al., 2014* | | CB15N *ftsW::ftsW*$_{A246T}$ |
| Strain, strain background (*Caulobacter crescentus*) | CB15N derivatives | This paper | | *Supplementary file 5* |
| Strain, strain background (*Escherichia coli*) | Rosetta(DE3)pLysS | Merck, Germany | Cat. #: 70956 | F⁻ *ompT hsdS*$_B$(r$_B$- m$_B$-) *gal dcm* (DE3) pLysSRARE (Cam$^R$) |
| Strain, strain background (*Escherichia coli*) | TOP10 | Invitrogen, Germany | Cat. #: C404003 | F⁻ *mcrA* Δ(*mrr-hsd*RMS-*mcr*BC) Φ80*lacZ*ΔM15 Δ*lacX74 rec*A1 *ara*D139 Δ(*ara leu*) 7697 *galU galK rps*L (StrR) *end*A1 *nup*G |
| Recombinant DNA reagent | pNPTS138 | M.R.K. Alley (unpublished) | | *sacB*-containing suicide vector used for homologous recombination, Kan$^R$ |
| Recombinant DNA reagent | pNPTS138 derivatives | This paper | | See *Supplementary file 6* |
| Recombinant DNA reagent | PPR9TT | *Santos et al., 2001* | | RK2-based replicating plasmid for the construction of lacZ fusions, Cam$^R$, Amp$^R$ |

*Continued on next page*

*Continued*

| Reagent type (species) or resource | Designation | Source or reference | Identifiers | Additional information |
|---|---|---|---|---|
| Recombinant DNA reagent | pPR9TT derivatives | This paper | | See *Supplementary file 6* |
| Recombinant DNA reagent | pTB146 | *Bendezú et al., 2009* | | Plasmid for overexpression of protein with N-terminal His$_6$-SUMO fusion, Amp$^R$ |
| Recombinant DNA reagent | pTB146 derivatives | This paper | | See *Supplementary file 6* |
| Sequence-based reagent | DNA oligonucleotides | This paper | | See *Supplementary file 7* |
| Sequence-based reagent | *amiR*$_{H26A,H133A,D143A}$ gene | Eurofins, Germany | Custom-synthesized | |
| Chemical compound, drug | ampicillin | Carl Roth, Germany | Cat. #: K029.3 | |
| Chemical compound, drug | aztreonam | Merck, Germany | Cat. #: A6848 | |
| Chemical compound, drug | cephalexin | Sigma Aldrich, Germany | Cat. #: C4895 | |
| Chemical compound, drug | chloramphenicol | Carl Roth, Germany | Cat. #: 3886.3 | |
| Chemical compound, drug | kanamycin | Carl Roth, Germany | Cat. #: T832.3 | |
| Chemical compound, drug | mecillinam | Merck | Cat. #: 33447 | |
| Chemical compound, drug | bocillin-FL | Invitrogen, Germany | Cat. #: B13233 | |
| Chemical compound, drug | D(+)-glucose | Carl Roth, Germany | Cat. #: 6887.1 | |
| Chemical compound, drug | ferrous sulfate chelate solution | Sigma Aldrich, Germany | Cat. #: F0518 | |
| Chemical compound, drug | fosfomycin MIC test stripes | Liofilchem, Italy | Cat. #: 92078 | |
| Chemical compound, drug | isopropyl-β-D-thiogalacto-pyranoside (IPTG) | Carl Roth, Germany | Cat. #: CN08.2 | |
| Chemical compound, drug | LB medium (Luria/Miller) | Carl Roth, Germany | Cat. #: X968.4 | |
| Chemical compound, drug | *N*-acetylglucosamine | Sigma Aldrich, Germany | Cat. #: A8625 | |
| Chemical compound, drug | *N*-acetylmuramic acid | Sigma Aldrich, Germany | Cat. #: A3007 | |
| Chemical compound, drug | nitrocefin | Merck, Germany | Cat. #: 484400 | |
| Chemical compound, drug | o-nitrophenyl-β-D-galactopyranosid | Carl Roth, Germany | Cat. #: CN22.1 | |
| Chemical compound, drug | Bacto Peptone | Thermo Fisher Scientific, Germany | Cat. #: 211677 | |
| Chemical compound, drug | Bacto Yeast Extract | BD Biosciences, Germany | Cat. #: 212750 | |
| Software, algorithm | Alphafold3 | *Abramson et al., 2024* | https://alphafoldserver.com | RRID:SCR_028034 |
| Software, algorithm | BLAST | *Altschul et al., 1990* | https://blast.ncbi.nlm.nih.gov/Blast.cgi | RRID:SCR_004870 |
| Software, algorithm | Fiji (2.14.0/1.54 f) | *Schindelin et al., 2012* | https://imagej.net/software/fiji | RRID:SCR_002285 |
| Software, algorithm | Oufti | *Paintdakhi et al., 2016* | https://oufti.org/ | RRID:SCR_016244 |
| Software, algorithm | SuperPlotsOfData web app | *Goedhart, 2021* | https://huygens.science.uva.nl/SuperPlotsOfData | |
| Software, algorithm | VolcaNoseR | *Goedhart and Luijsterburg, 2020* | https://goedhart.shinyapps.io/VolcaNoseR | |

## Experimental models

The bacterial strains used in this study are derivatives of *Caulobacter crescentus* NA1000 (CB15N) (*Poindexter, 1964*), *Escherichia coli* TOP10 (Thermo Fisher Scientific, Germany) or *Escherichia coli* Rosetta(DE3)pLysS (Merck, Germany). The sequences of all genes and proteins investigated are based on the genome sequence of *C. crescentus* NA1000 (*Nierman et al., 2001*).

## Cultivation of bacterial strains

*C. crescentus* NA1000 (CB15N) and its derivatives were grown at 28 °C in peptone-yeast-extract (PYE) medium (*Poindexter, 1964*), double-concentrated PYE (2xPYE) medium, or M2 salts-glucose (M2G) minimal medium (*Hottes et al., 2004*) under aerobic conditions, shaking at 210 rpm. To prepare PYE plates, liquid medium was solidified by the addition of 1.5% agar. To ensure the maintenance of integrative plasmids, media were supplemented with kanamycin (5 µg/mL for liquid cultures, 25 µg/mL for plates). To counter-select against cells carrying the *sacB* marker, solid PYE medium was supplemented with 3% D-sucrose. *E. coli* TOP10 (Invitrogen, Germany) was used for cloning purposes, whereas *E. coli* Rosetta(DE3)pLysS (Novagen, Germany) was used for protein overproduction. Cells were grown aerobically at 37 °C in LB Broth or on LB Agar (Carl Roth, Germany), supplemented when appropriate with antibiotics at the following concentrations (µg/mL; liquid/solid): kanamycin (30/50), ampicillin (200/200), chloramphenicol (20/30).

## Plasmid and strain construction

The bacterial strains, plasmids, and oligonucleotides used in this study are listed in *Supplementary files 5–7*. *C. crescentus* was transformed by electroporation (*Ely, 1991*). Gene replacement was achieved by double homologous recombination using the counter-selectable *sacB* marker (*Thanbichler and Shapiro, 2006*). Proper chromosomal integration or gene replacement was verified by colony PCR.

## Antibiotic susceptibility assay

Overnight cultures were serially diluted ($10^{-2}$ to $10^{-7}$) in PYE medium and spotted on PYE plates containing ampicillin (10–50 µg/mL) or aztreonam (95–105 µg/mL) at the indicated concentrations. When appropriate, the media were additionally supplemented with 0.3% GlcNAc or 100 µg/mL MurNAc. The plates were incubated for three days at room temperature (25 °C) and imaged using a ChemiDocTM XRS +Imager (Biorad, Germany) and Image Lab software (version 5.0) (Biorad, Germany). To account for the high instability of ampicillin in aqueous solution, ampicillin stocks were prepared freshly before every experiment.

Resistance against fosfomycin was assessed using Fosfomycin MIC Test Strips (Liofilchem, Italy). Overnight cultures were diluted with PYE medium to an $OD_{600}$ of 0.1. Subsequently, 100 µL of the suspensions were added to 20 mL of sterile liquid PYE agar that had been cooled to slightly above the gelation temperature (~45 °C). After mixing, the suspension was immediately poured into cultivation dishes. After solidification of the medium, the plates were briefly dried, and the test strips were applied on their top. Subsequently, the plates were incubated for three days at 28 °C and imaged using a ChemiDocTM XRS+Imager and Image Lab 5.0 software (Bio-Rad Laboratories, Germany).

## Light microscopy

Unless indicated otherwise, cells were grown in PYE medium, transferred to 1% agarose pads, and imaged with a Zeiss Axio Imager.Z1 microscope equipped with a Plan Apochromat 100x/1.45 Oil DIC and a 100x/1.40 Oil Ph3 M27 phase contrast objective. Images were recorded with a pco.edge 4.2 sCMOS camera (PCO). Images were recorded with VisiView 4.0.0.14 (Visitron Systems, Germany) and processed with Fiji 1.53 (*Schindelin et al., 2012*). Cell length measurements were performed with Oufti (*Paintdakhi et al., 2016*), using phase-contrast images as an input.

## Microfluidics-based ampicillin sensitivity assay

Microfluidic-based imaging was performed using a CellASIC ONIX B04A plate for bacterial cells (Merck, Germany) that was connected to a CellASIC ONIX2 microfluidic system (Merck, Germany) and imaged with a Zeiss Axio Observer.Z1 (see above) in a heatable enclosure adjusted to 28 °C. Exponentially growing *C. crescentus* wild-type and Δ*amiR* cells were diluted to an $OD_{600}$ of 0.2 and flushed into

different chambers of the microfluidic plate. They were then first incubated for 3 hr with a constant flow of PYE medium and then for 5 hr with a constant flow of PYE medium containing 20 μg/mL ampicillin at a pressure of 5 kPa. DIC images were taken at 5 min intervals to follow cell growth and division.

## β-lactamase activity assay

β-lactamase activity was assessed using the chromogenic cephalosporin derivative nitrocefin (Merck, Germany) as a substrate (*O'Callaghan et al., 1972*). Cells were grown to exponential phase in 3 mL PYE medium, harvested by centrifugation for 5 min at 7000×*g* and room temperature, and resuspended in 1 mL PBS. After transfer of the suspensions into a 24-well polystyrene microplate (Carl Roth, Germany), the reactions were started by the addition of 100 μL nitrocefin (1 mg/mL). The hydrolysis of nitrocefin was then monitored by measuring the increase in absorbance at 482 nm every 2 min over a period of 180 min at 30 °C using an Epoch 2 microplate reader (BioTek, California, United States).

## Identification of ampicillin target proteins

*C. crescentus* CS606 (Δ*blaA*) cells were grown to exponential phase in 100 mL PYE medium, harvested, washed with PBS, and lysed by three passages through a French press at 10,000 psi. Insoluble material, including the membrane fraction, was collected by centrifugation for 15 min at 21,000×*g* (4 °C). After resuspension of the pellet in 2 mL of PBS, 100 μL aliquots were incubated with 0, 1, 10, or 100 μg/mL ampicillin, 5 μg/mL cefalexin or 100 μg/mL mecillinam for 2 hr at room temperature. The membranes were collected again by centrifugation (1 min, 13,000×*g*, room temperature), washed twice with PBS, and resuspended in a final volume of 100 μL PBS containing 5 μg/mL Bocillin-FL (Thermo Fisher Scientific, Germany). After 10 min of incubation, the membranes were collected by centrifugation (1 min, 13,000×*g*, room temperature), washed with PBS, dissolved in 30 μL of 2 x SDS sample buffer (0.125 M Tris Base, 20% [v/v] glycerol, 4% [w/v] SDS, 200 mM DTT, 0.01% [w/v] bromophenol blue, pH 6.8), heated for 10 min at 90 °C and loaded onto an 8% SDS-polyacrylamide gel. After separation by electrophoresis, proteins labeled with Bocillin-FL were detected with a ChemiDoc MP imager (Bio-Rad Laboratories, Germany) using a 526 nm short-pass filter.

## β-galactosidase assay

Cultures were grown to exponential phase in PYE medium. After determining the cell densities (OD$_{600}$), the cells were harvested by centrifugation for 7 min at 12,000×*g* and 4 °C and resuspended in 1 mL of Z-buffer (60 mM Na$_2$PO$_4$, 40 mM NaH$_2$PO$_4$, 10 mM KCl, 1 mM MgSO$_4$, 50 mM β-mercaptoethanol, pH 7.0). Following the addition of 100 μL chloroform and 0.1% SDS, the suspensions were briefly agitated and then incubated for 7 min at room temperature to permeabilize the cells. 500 μL samples of the suspensions were mixed with 500 μL of Z-buffer. The reactions were then started by the addition of 200 μL *o*-nitrophenyl-β-D-galactopyranoside (4 mg/mL), incubated at room temperature until the mixtures turned yellow, and stopped by the addition of 500 μL 1 M Na$_2$CO$_3$. Cell remnants were pelleted by centrifugation, and the extinction of the supernatant was measured photometrically at 550 nm and 420 nm. β-galactosidase activities were then calculated as follows: Activity (in Miller units)=$1000*(OD_{420}–1.75*OD_{550})/(t*V*OD_{600})$, where *t* is the time of incubation (min) and *V* the culture volume used for the reaction (mL).

## Protein purification

To purify **AmiR and AmiR\*** (H26A/H133A/D143A), *E. coli* Rosetta(DE3)pLysS was transformed with plasmids encoding His$_6$-SUMO-tagged derivatives (*Marblestone et al., 2006*) of the proteins (pPR029 or pPR041) and grown in LB medium to an OD$_{600}$ of 0.4. After reduction of the growth temperature to 18 °C, the cultures were supplemented with isopropyl-β-D-thiogalactopyranoside (IPTG) to a final concentration of 1 mM and incubated further overnight. The cells were harvested by centrifugation (12 min, 5000×*g*, 4 °C), washed in buffer A (50 mM Tris-HCl, 500 mM NaCl, 10 mM imidazole, 10% glycerol, pH 7.0), and stored at –80 °C until needed. After thawing, the cells were resuspended in buffer A containing 100 μg/mL phenylmethylsulfonyl fluoride (PMSF) and 10 μg/mL DNaseI and lysed by three passages through a French press at a pressure of 10,000 psi. The lysate was cleared by centrifugation (30 min, 25,000×*g*, 4 °C) and loaded onto a His-Trap HP 5 mL Ni-NTA affinity column (Cytiva, Germany) equilibrated with buffer A. After washing of the column with buffer A, protein was eluted at a flow rate of 2 mL/min with a linear gradient of imidazole generated by mixing buffer A

and buffer B (50 mM Tris-HCl, 500 mM NaCl, 250 mM imidazole, 10% glycerol, pH 7.0). The peak fractions were pooled, supplemented with SUMO protease (His$_6$-Ulp) (*Marblestone et al., 2006*) at a molar ratio of 1:1000 relative to the purified protein, and dialyzed overnight at 4 °C against buffer C (50 mM Tris-HCl, 500 mM NaCl, 10% glycerol, 1 mM dithiothreitol, pH 7.0). The solution was then loaded again onto a His-Trap HP 5 mL column to remove His$_6$-SUMO and His$_6$-Ulp. Flow-through fractions containing the protein of interest were collected, concentrated using Amicon Ultra centrifugation filters (10 kDa MWCO; Merck, Germany), loaded onto a Superdex 75 GL10/300 size-exclusion column (Cytiva, Germany) equilibrated with buffer D (50 mM HEPES-NaOH, 500 mM NaCl, pH 7.0) and eluted at a flow rate of 10 mL/min. Fractions containing the protein of interest at high concentration and purity were combined, concentrated, snap-frozen in liquid nitrogen and stored at –80 °C until further use.

To purify **NagZ and NagZ\*** (D259A), His$_6$-SUMO-tagged derivatives of the proteins were overproduced in *E. coli* Rosetta(DE3)pLysS carrying suitable plasmids (pPR102 or pPR107) as described above. The fusion proteins were then purified on a His-Trap HP 5 mL column, cleaved with SUMO protease, and separated from His$_6$-SUMO and His$_6$-Ulp by a second affinity purification step essentially as described for AmiR, with the exceptions that buffers A and B contained 300 mM instead of 500 mM NaCl and buffer C contained 150 mM instead of 500 mM NaCl. Flow-through fractions from the His-Trap HP column that contained the protein of interest at high concentration and purity were concentrated, snap-frozen in liquid nitrogen and stored at –80 °C.

## Muropeptide analysis

The *C. crescentus* wild type and the Δ*amiR* mutant were grown to stationary phase (OD$_{600}$ ≈ 1.2) in 400 mL PYE medium and harvested by centrifugation at 16,000×*g* and 4 °C for 30 min. The pelleted cells were resuspended in 6 mL of ice-cold double-ionized water ($_{dd}$H$_2$O) and added dropwise to 6 mL of a boiling solution of 8% sodium dodecylsulfate (SDS), which was vigorously stirred. The suspension was boiled for an additional 30 min, and double-deionized water was added regularly to maintain a constant volume. Subsequently, the lysate was cooled to room temperature and stored at 4 °C until further use. Peptidoglycan was isolated following previously described procedures (*Glauner, 1988*; *Takacs et al., 2010*) and digested with the muramidase cellosyl (kindly provided by Hoechst, Frankfurt, Germany). The resulting muropeptides were reduced with sodium borohydride and separated by HPLC as described previously (*Takacs et al., 2010*). The identity of eluted fragments was assigned based on the retention times of known muropeptides (*Takacs et al., 2010*).

## AmiR and NagZ activity assays

To test the activity of AmiR towards anhydro-muropeptides, peptidoglycan from *E. coli* D456 (*Edwards and Donachie, 1993*), which lacks three major DD-carboxypeptidases and thus has elevated levels of pentapeptides, was treated with the soluble lytic transglycosylase Slt70 from *E. coli* (*Banzhaf et al., 2020*; *Betzner and Keck, 1989*). The products were incubated with wild-type AmiR or a catalytically inactive AmiR derivative (AmiR\*) in a total volume of 50 µl in buffer H (20 mM HEPES-NaOH, 50 mM NaCl, 1 mM MgCl$_2$, pH 7.5) at 37 °C for 16 hr in a thermal shaker set to 900 rpm. The reactions were stopped by incubation at 100 °C for 10 min in a dry-bed heater. After clearance of the mixtures by centrifugation at 17,000×*g* for 10 min, the supernatants were acidified to pH 4.0 – pH 4.5 with 20% phosphoric acid. Subsequently, the reaction products were separated by HPLC as described previously (*Izquierdo-Martinez et al., 2023*). The activity of AmiR towards muropeptides with reducing ends was tested in a similar manner. However, peptidoglycan from *E. coli* D456 was treated with the muramidase cellosyl from *Streptomyces coelicolor* (*Bräu et al., 1991*). After incubation with the enzyme, heat inactivation and centrifugation, the supernatants were mixed with an equal volume of 0.5 M sodium borate (pH 9.0), followed by reduction of the muropeptides with ~1 mg of sodium borohydride. The samples were then acidified to pH 4.0–pH 4.5 with 20% phosphoric acid and analyzed by HPLC as described previously (*Glauner, 1988*).

The enzymatic activity of NagZ towards anhydro-muropeptides or reducing muropeptides was analyzed as described for AmiR, but with the following modifications. Peptidoglycan from *E. coli* CS703-1 (*Meberg et al., 2001*), which lacks all major DD-carboxypeptidases, was treated either with the lytic transglycosylase MltA from *E. coli* (*Lommatzsch et al., 1997*) to produce anhydro-muropeptides or with the muramidase cellosyl to produce muropeptides with reducing ends. The

lytic products were incubated with NagZ or a catalytically inactive NagZ variant (NagZ*) (10 μM) and processed as described above.

To determine the activity of AmiR towards undigested peptidoglycan, sacculi isolated from *E. coli* D456 (*Glauner, 1988*; *Takacs et al., 2010*) were incubated with AmiR or AmiR* (10 μM) in buffer H at 37 °C for 16 hr in a thermal shaker set to 900 rpm. The reactions were stopped by heating at 100 °C for 10 min in a dry-bed heater. Subsequently, the sacculi were digested with cellosyl overnight. After heat inactivation of cellosyl at 100 °C for 10 min and clearance of the mixtures by centrifugation, the reaction products were reduced with borohydride. The samples were then acidified to pH 4.0–pH 4.5 with 20% phosphoric acid and analyzed by HPLC as described previously (*Glauner, 1988*).

In the above analyses, disaccharide-containing muropeptides and anhydro-muropeptides were identified based on their known retention times and peak patterns (*Glauner, 1988*). The identities of the Glc*N*Ac–anhMur*N*Ac disaccharide generated by AmiR and the anhMur*N*Ac-peptides generated by NagZ were determined by HPLC separation of the reaction products and mass spectrometric analysis of the different peak fractions (*Figure 2—figure supplement 4*).

To assess the activity of AmiR against anhydro-muropeptide species lacking the Glc*N*Ac moiety, a solution of purified Glc*N*Ac–anhMur*N*Ac-tetrapeptide (5 μL) was incubated with purified NagZ (10 μM) in 90 μL reaction buffer (50 mM HEPES, 200 mM NaCl, 10% [v/v] glycerol, pH 7.2) with shaking for 16 hr at 37 °C. Subsequently, the reactions were supplemented with AmiR or AmiR* (10 μM) and incubated for another 4 hr. The mixtures were acidified by the addition of 10 μL of 10% [v/v] formic acid, and precipitated protein was removed by centrifugation for 10 min at 30,130×g. The cleared solutions were then subjected to liquid chromatography-mass spectrometry (LC-MS) analysis for semi-quantitative determination of the reaction products, as described under 'Metabolomics analysis.' The compounds of interest were identified based on their theoretical mass-to-charge (m/z) ratios.

## Metabolomics analysis

Anhydro-muropeptides were quantified essentially as described previously (*Jacobs et al., 1994*; *Simpson et al., 2023*). In brief, cultures were grown overnight in 10 mL PYE medium (OD$_{600}$=1.2) and harvested by centrifugation for 10 min at 21,000×g and 4 °C. The pelleted cells were washed three times with 0.9% NaCl, resuspended to a final volume of 100 μL in 0.9% NaCl and heated for 7 min at 98 °C. The suspension was cleared by centrifugation for 15 min at 21,000×g. The supernatant was isolated, centrifuged again to remove residual cell debris, and then subjected to liquid chromatography-mass spectrometry (LC-MS) analysis for semi-quantitative determination of PG recycling or biosynthesis intermediates. The chromatographic separation was performed on an Agilent Infinity II 1290 HPLC system using a ZicHILIC SeQuant column (150×2.1 mm, 3.5 μm particle size, 100 Å pore size) connected to a ZicHILIC guard column (20×2.1 mm, 5 μm particle size) (Merck KgAA) at 25 °C and at a constant flow rate of 0.3 mL/min, with mobile phase A being 0.1% formic acid in 99:1 water:acetonitrile (Honeywell, Morristown, NJ, USA) and phase B being 0.1% formic acid in 99:1 acetonitrile:water (Honeywell, Morristown, NJ, USA). The injection volume was 5 μL. The mobile phase profile comprised the following steps and linear gradients: 0–1 min constant at 80% B; 1–10 min from 80 to 10% B; 10–12 min constant at 10% B; 12–12.1 min from 10 to 80% B; 12–14 min constant at 80% B. An Agilent 6546 A QTOF mass spectrometer was used in high-resolution, positive mode with a mass range of 70–1700 m/z and a scan rate of 1.5 spectra/second. An electrospray ionization source was used at the following conditions: ESI spray voltage 4000 V, nozzle voltage 500 V, sheath gas 250 ° C at 12 L/min, nebulizer pressure 60 psig, and drying gas 100 °C at 11 L/min. Compounds were identified based on their accurate mass and retention time, applying an extraction tolerance of 5 ppm. Chromatograms were integrated using MassHunter software (Agilent, CA, USA). Relative abundance was determined based on the peak area.

To determine the cytoplasmic amounts of the PG precursors UDP-Glc*N*Ac and UDP-Mur*N*Ac, the strains to be analyzed were cultivated overnight in PYE medium, followed by determination of their optical densities (OD$_{600}$). Subsequently, 1 mL samples of the cultures were mixed with 1 mL of 70% LC-MS-Grade methanol (Merck, Germany) and subjected to centrifugation at 13,000×g and –9 °C for 10 min. To extract the endometabolome, the pelleted cells were mixed with 100 μL per OD$_{600}$ unit of extraction fluid (50% LC-MS-Grade methanol, 5 mM Tris, 0.5 mM EDTA) and the same amount of chloroform, followed by incubation at 4 °C for 2 hr. After separation of the phases by centrifugation (max. speed, –9 °C, 10 min), the upper phase was isolated, filtered using PTFE membrane filters

(0.2 μm pore size) (Sartorius, Germany), and subjected to LC-MS analysis. The chromatographic separation was performed on an Agilent Infinity II 1290 HPLC system using a SeQuant ZIC-pHILIC column (150×2.1 mm, 5 μm particle size, peek coated, Merck) connected to a guard column of similar specificity (20×2.1 mm, 5 μm particle size, Phenomenex) at 40 °C and at a constant flow rate of 0.1 mL/min, with mobile phase A composed of 10 mM ammonium acetate in water (pH 9) supplemented with medronic acid to a final concentration of 5 μM and mobile phase B composed of 10 mM ammonium acetate in 90:10 acetonitrile:water (pH 9). The injection volume was 5 μL. The mobile phase profile comprised the following steps and linear gradients: 0–1 min constant at 75% B; 1–6 min from 75 to 40% B; 6–9 min constant at 40% B; 9–9.1 min from 40 to 75% B; 9.1–20 min constant at 75% B. An Agilent 6546 A QTOF mass spectrometer was used in high-resolution, negative mode with a mass range of 50–1700 m/z and a scan rate of 1.5 spectra/second. An electrospray ionization source was used at the following conditions: ESI spray voltage 4000 V, nozzle voltage 500 V, sheath gas 300 °C at 12 L/min, nebulizer pressure 20 psig, and drying gas 150 °C at 11 L/min. Compounds were identified based on their accurate mass and retention time, applying an extraction tolerance of 5 ppm. Chromatograms were integrated using MassHunter software (Agilent, Santa Clara, CA, USA). Relative abundance was determined based on the peak area.

## Whole-cell proteomics

Cells were grown to exponential phase in 10 mL PYE medium, harvested by centrifugation (15,000×g, 10 min, 4 °C), washed three times with ice-cold PBS, and resuspended in 300 μL of lysis buffer (2% sodium lauroyl sarcosinate [SLS], 100 mM ammonium bicarbonate). The samples were then heated for 10 min at 90 °C, followed by sonication with a VialTweeter Sonicator (Hielscher Ultrasonics, Germany). Proteins were reduced with 5 mM Tris (2-carboxyethyl) phosphine (Thermo Fisher Scientific, Germany) at 90 °C for 15 min and alkylated using 10 mM iodoacetamide (Sigma Aldrich, Germany) at 20 °C for 30 min in the dark. Subsequently, they were precipitated by the addition of a sixfold excess of ice-cold acetone and incubation for 2 hr at –20 °C, followed by two washing steps with methanol. After drying, the proteins were reconstituted in 0.2% SLS and their amount was determined using a bicinchoninic acid (BCA) protein assay (Thermo Scientific, Germany). Subsequently, 50 μg protein were subjected to tryptic digestion by incubation with 1 μg of trypsin (Serva) in 0.5% SLS at 30 °C overnight. After digestion, SLS was precipitated by adding a final concentration of 1.5% trifluoroacetic acid (TFA, Thermo Fisher Scientific, Germany). The peptides were desalted using CHROMABOND C18 solid phase extraction cartridges (Macherey-Nagel, Germany), which were prepared by the addition of acetonitrile and subsequent equilibration with 0.1% TFA. They were then loaded on the equilibrated cartridges, washed with a mixture of 5% acetonitrile and 0.1% TFA, and finally eluted with a mixture of 50% ACN and 0.1% TFA. After drying, the peptides were reconstituted in 0.1% TFA and then analyzed using liquid-chromatography-mass spectrometry on an Exploris 480 instrument connected to an UltiMate 3000 RSLCnano HPLC system and a Nanospray Flex ion source (all Thermo Scientific, Germany). The following gradient was used for separation: 94% solvent A (0.15% formic acid) and 6% solvent B (99.85% acetonitrile, 0.15% formic acid) to 25% solvent B over 40 min, followed by an increase to 35% solvent B over 20 min at a flow rate of 300 nl/min. Peptides were ionized at a spray voltage of 2.3 kV, with the ion transfer tube temperature set at 275 °C. 445.12003 m/z was used as internal calibrant.

MS raw data were acquired on an Orbitrap Exploris 480 instrument (Thermo Scientific, Germany) in data-independent acquisition (DIA) mode. The funnel RF level was set to 40. For DIA experiments, full MS resolution was set to 120,000 at m/z 200. The AGC target value for fragment spectra was set at 3000%. Forty-five windows of 14 Da were used with an overlap of 1 Da between m/z 320–950. Resolution was set to 15,000 and IT to 22ms. A stepped HCD collision energy of 25, 27.5, 30% was used. MS1 data were acquired in profile, MS2 DIA data in centroid mode.

The analysis of DIA data was performed with DIA-NN version 1.8 (*Demichev et al., 2020*), based on a protein database for *Caulobacter crescentus* NA1000 retrieved from UniProt (*Bateman et al., 2025*) to generate a dataset-specific spectral library. The DIA-NN suite was used to perform noise interference correction (mass correction, RT prediction, and precursor/fragment co-elution correlation) and peptide precursor signal extraction on the DIA-NN raw data. The following parameters were used: Full tryptic digest was allowed with two missed cleavage sites, oxidized methionines, and carbamidomethylated cysteines. 'Match between runs' and 'Remove likely interferences' were enabled. The precursor FDR was set to 1%. The neural network classifier was set to the single-pass mode,

and protein inference was based on genes. The quantification strategy was set to any LC (high accuracy). Cross-run normalization was set to RT-dependent. Library generation was set to smart profiling. DIA-NN outputs were further evaluated using the SafeQuant script (*Ahrné et al., 2013*; *Glatter et al., 2012*), modified to process DIA-NN outputs. A detailed summary of the proteomics data obtained is provided in *Supplementary file 4*.

### Bioinformatic analysis

Protein similarity searches were performed with BLAST (*Altschul et al., 1990*), using the BLAST server of the National Institutes of Health (https://blast.ncbi.nlm.nih.gov/Blast.cgi). Protein structures were predicted with Alphafold 3 (*Abramson et al., 2024*) and visualized using PyMOL v2.6.0a0 (Schrödinger LLC). Proteomics data were visualized with VolcaNoseR (*Goedhart and Luijsterburg, 2020*).

### Quantification and statistical analysis

Details on the number of replicates and the sample sizes are given in the figure legends. Datasets were displayed using the SuperPlotsOfData web application (*Goedhart, 2021*), and an unpaired two-sided Welch's *t*-test implemented in the application was used to assess the statistical significance of differences between strains or conditions. Standard deviations were calculated in Microsoft Excel 2019.

### Availability of biological material

The plasmids and strains used in this study are available from the corresponding author upon request.

## Acknowledgements

We thank Julia Rausch for excellent technical assistance and Daniela Vollmer for the purification of peptidoglycan.

## Additional information

### Funding

| Funder | Grant reference number | Author |
| --- | --- | --- |
| Max Planck Society | Max Planck Fellowship | Martin Thanbichler |
| Biotechnology and Biological Sciences Research Council | BB/W013630/1 | Waldemar Vollmer |
| Marburg University | Core funding | Martin Thanbichler |

The funders had no role in study design, data collection and interpretation, or the decision to submit the work for publication.The funders had no role in study design, data collection and interpretation, or the decision to submit the work for publication.

### Author contributions

Pia Richter, Conceived the study; Generated plasmids and strains,; Performed the in vivo experiments; Purified the proteins; Prepared the samples for the proteomics and metabolomics analyses; Performed the bocillin labeling experiments; Analyzed, visualized and curated the data; Wrote the manuscript, with input from all other authors; Anna Merz, Generated plasmids and strains; Jacob Biboy, Conducted the enzyme assays and muropeptide analyses; Nicole Paczia, Performed the metabolomics analyses; Timo Glatter, Performed the proteomics analyses; Jared Ng, Prepared GlcNAc-anhMurNAc-tetrapeptide; Waldemar Vollmer, Supervised the enzyme assays and muropeptide analyses; Secured funding; Analyzed the data; Revised the manuscript; Martin Thanbichler, Conceived and supervised the study; Secured funding; Analyzed and visualized the data; Wrote the manuscript, with input from all other authors

## Author ORCIDs
Pia Richter ⓘ https://orcid.org/0009-0006-1332-9149
Jacob Biboy ⓘ https://orcid.org/0000-0002-1286-6851
Martin Thanbichler ⓘ https://orcid.org/0000-0002-1303-1442

Reviewer #1 (Public review): https://doi.org/10.7554/eLife.109465.3.sa1
Author response https://doi.org/10.7554/eLife.109465.3.sa2

---

# Additional files

## Supplementary files
Supplementary file 1. Proteins found to be differentially accumulated in Δ*regX* cells compared to wild-type cells. The table lists the ORF numbers and predicted functions of the numbered proteins in *Figure 8—figure supplement 1C*.

Supplementary file 2. Muropeptide composition of peptidoglycan isolated from stationary *C. crescentus* wild-type and Δ*amiR* cells. The table gives the relative abundance of the indicated muropeptide species, calculated from the areas of the corresponding peaks in the HPLC chromatograms from *Figure 9—figure supplement 1*.

Supplementary file 3. Overview of the muropeptide species identified in peptidoglycan from stationary *C. crescentus* wild-type and Δ*amiR* cells. The table summarizes the relative abundance of different muropeptide species, calculated from the values listed in *Supplementary file 2*.

Supplementary file 4. Summary of the proteomics data obtained in this study. The spreadsheets show the data underlying the volcano plots in *Figure 5D*, *Figure 8D*, *Figure 8—figure supplement 1C* and *Figure 8—figure supplement 2*.

Supplementary file 5. Strains used in this study.

Supplementary file 6. Plasmids used in this study.

Supplementary file 7. Oligonucleotides used in this study.

MDAR checklist

Source data 1. Source data underlying the figures in this paper.

## Data availability
The proteomics data generated in this study have been deposited to the ProteomeXchange Consortium via the PRIDE partner repository (*Perez-Riverol et al., 2025*) under the dataset identifier PXD069004. All other data supporting the findings of this study are included in the main text, the supplementary material and the source data file.

The following dataset was generated:

| Author(s) | Year | Dataset title | Dataset URL | Database and Identifier |
|---|---|---|---|---|
| Glatter T | 2025 | Peptidoglycan recycling is critical for cell division, cell wall integrity and β-lactam resistance in *Caulobacter crescentus* | https://www.ebi.ac.uk/pride/archive/projects/PXD069004 | PRIDE, PXD069004 |

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
