## [Editor Report · eLife Assessment]

This manuscript presents a **valuable** investigation of the peptidoglycan (PG) recycling pathway in *Caulobacter crescentus*. The authors showed that PG recycling in *C. crescentus* is essential not only for β-lactam (ampicillin) resistance but also for cell morphology, efficient division, and overall fitness. The study is comprehensive and **compelling**.

---

## [Referee Report · Reviewer #1 (Public review)]

Summary:

In their manuscript, Richter and colleagues comprehensively investigate the cell wall recycling pathway in the model alphaproteobacterium Caulobacter crescentus using biochemical, imaging, and genetic approaches. They clearly demonstrate that this organism encodes a functional peptidoglycan recycling pathway and demonstrate the activities of many enzymes and transporters within this pathway. They leverage imaging and growth assays to demonstrate that mutants in peptidoglycan recycling have varying degrees of beta-lactam sensitivity as well as morphological and cell division defects. They propose that, rather than impacting the levels or activity of the major beta-lactamase, BlaA, defects in PG recycling lead to beta-lactam sensitivity by limiting the availability of new cell wall precursors. The findings will be of interest to those in the field of bacterial cell wall biochemistry, antibiotics and antibiotic resistance, and bacterial morphogenesis.

Strengths:

Overall the manuscript is laid out logically, and the data are comprehensive, quantitative, and rigorous. The mutants and their phenotypes will be a valuable resource for Caulobacter researchers, and the findings may be relevant to cell wall recycling in other organisms.

Weaknesses:

No major weaknesses are noted.

Comments on revisions:

The authors addressed all of our concerns with the initial submission.

---

## [Author Response]

The following is the authors’ response to the original reviews.

**Public Reviews:**

**Reviewer #1 (Public review):**
Summary:In their manuscript, Richter and colleagues comprehensively investigate the cell wall recycling pathway in the model alphaproteobacterium Caulobacter crescentus using biochemical, imaging, and genetic approaches. They clearly demonstrate that this organism encodes a functional peptidoglycan recycling pathway and demonstrate the activities of many enzymes and transporters within this pathway. They leverage imaging and growth assays to demonstrate that mutants in peptidoglycan recycling have varying degrees of beta-lactam sensitivity as well as morphological and cell division defects. They propose that, rather than impacting the levels or activity of the major beta-lactamase, BlaA, defects in PG recycling lead to beta-lactam sensitivity by limiting the availability of new cell wall precursors. The findings will be of interest to those in the field of bacterial cell wall biochemistry, antibiotics and antibiotic resistance, and bacterial morphogenesis.Strengths:Overall, the manuscript is laid out logically, and the data are comprehensive, quantitative, and rigorous. The mutants and their phenotypes will be a valuable resource for Caulobacter researchers.

Thank you for this positive evaluation. Previous work has mostly focused on the role of PG recycling in the regulation of *ampC* expression. However, our study and recent work in *A. tumefaciens* (Gilmore & Cava, 2022) and *C. crescentus* (Modi et al, 2025) demonstrates that β-lactam resistance is heavily influenced by PG recycling and the metabolic state of the cell, even in the presence of high levels of β-lactamase activity. It is likely that these effects are not limited to the two alpha­proteo­bacterial species investigated to date but may be more widely applicable. Therefore, we believe that our results are relevant beyond the *Caulobacter* field and may help to stimulate similar analyses in other, medi­cally more relevant species.

Weaknesses:

The only major missing piece is the complementation of mutants to demonstrate that loss of the targeted gene is responsible for the observed phenotypes.

In our initial manuscript, we showed that the replacement of the native AmiR and NagZ genes with mutant alleles encoding catalytically inactive variants of the two proteins gave rise to the same pheno­types as gene deletions. This finding indicates that the defects observed were due to the loss of AmiR or NagZ activity, respectively. To rule out artifacts from polar effects, we have now also conducted the requested complementation analysis for the Δ*ampG*, Δ*amiR* and Δ*nagZ* mutants. The results obtained show that deletion mutants carrying an ectopically expressed wild-type gene copy behave essentially like the wild-type strain, thereby verify­ing the validity of our conclusions (new Figure 4-figure supple­ment 1).

**Reviewer #2 (Public review):**
Summary:Pia Richter et al. investigated the peptidoglycan (PG) recycling metabolism in the alpha-proteobacterium Caulobacter crescentus. The authors first identified a functional recycling pathway in this organism, which is similar to the Pseudomonas route, and they characterized two key enzymes (NagZ, AmiR) of this pathway, showing that AmiR differs in specificity from the AmpD counterpart of *E. coli*. Further, they studied the effects of deletions within the PG recycling pathway (ampG, amiR, nagZ, sdpA, blaA, nagA1, nagA2, amgK, nagK mutants), showing filamentation and cell widening, thereby revealing a link between PG recycling and cell division. Finally, they provide a link between PG recycling and beta-lactam sensitivity in C. crescents that is not caused by activation of a beta-lactamase, but rather is a result of reduced supply of PG building blocks increasing the sensitivity of penicillin-binding proteins.Strengths:This work adds to the understanding of the role of PG recycling in alpha-proteobacteria, which significantly differ in their mode of cell wall growth from the better studied gamma-proteobacteria.

Thank you for pointing out the relevance of our work. As mentioned above, we believe that our work goes beyond understanding the PG recycling pathway in alphaproteobacteria. Importantly, together with previous work, our results demonstrate a so-far largely neglected critical role of PG recycling in β-lactam resistance that goes beyond the mere regula­tion of β-lactamase gene expression. It will be interesting to determine the conservation of this phenomenon among other bacteria and to see whether blocking PG recycling could represent a potential strategy to combat β-lactam resistant pathogens.

Weaknesses:The findings are not entirely novel as recent studies by Modi et al. 2025 mBio (studying C. crescentus) and Gilmore & Cava 2022 Nat. Commun. (studying Agrobacterium tumefaciens) came to similar conclusions.

Gilmore & Cava have made the seminal finding that blocking anhydro-muropeptide import affects cell wall integrity in a manner that is partly independent of its effect on *ampC* expression. We now extend this finding by investigating various critical steps in the PG recycling pathway of *C. cres­centus*, a species lacking an AmpC homolog. Interestingly, by characterizing a variety of different mutants, we show that the morphol­ogical and ampicillin resistance defects they exhibit are not strictly con­nected and vary substantially between strains, suggesting that different steps in PG recycling differ in their importance for cellular fitness and cell wall integrity. This finding suggests that the phenotypes observed are not simply determined by the efficiency of PG recycling but likely result from a combination of factors. Based on the results obtained, we propose a model that highlights the different factors that may be at play and suggests a mechanism explaining their effects on β-lactam resistance and cell division. Our findings partly overlap with the recent study by Modi et al., but there are various points in which we disagree with their findings and conclusions. The need to rigorously validate our differing results led to a signi­ficant delay in the submission of our manuscript.

**Reviewer #1 (Recommendations for the authors):**
Major CommentGenetic complementation is lacking for deletion mutants throughout. Could you please provide complemented strains for mutants in key figures where deletion phenotypes are central to the conclusions (e.g., Figure 4 and related supplements).

As explained above, we have not performed the requested comple­mentation experiments and included the data as Figure 4-figure supplement 1.

Other minor comments:(1) Figure 1(a) This is a busy schematic; please consider visually separating PG biosynthesis vs. recycling (e.g., a faint divider line or shaded boxes).

We have now simplified the schematic and visually separated the PG recycling and de novo biosyn­thesis pathways.

(b) Please label "Fructose-6-phosphate" and "Glucosamine-6-phosphate (GlcN-6-P)" on the figure, since they are referenced in the caption (line 1410).

The symbols for fructose, glucosamine and phosphate are given in the legend on the right. For consistency, we would therefore prefer not to additionally label these compounds in the figure.

(c) Define all abbreviations in the caption: CM, GTase, TPase; and clarify the legend conventions (e.g., bold vs. regular font; red vs. black text).

The structure of PG and the different lytic enzymes have now been removed from Figure 1. All remaining abbreviations have now been defined in the legend.

(2) Figure 2 - Figure Supplement 2(a) Panel B: Please include the full chromatogram (it seems to be cropped at 10 min?). For AmiR in particular, it is important to show there are no nearby peaks at earlier retention times (eg GlcNAc).

The region before 10 min is cropped in many published muropeptide profiles because the peaks contained in it are known to correspond to salts, i.e., borate from the reduction step and phos­phate, which are poorly retained on the C18 column (Figure 2–figure supplement 2). As the reviewer stated, free Glc*N*Ac would elute in this region and would not be recognized if it were produced by AmiR. However, AmiR cleaves free anhydro-muropeptides between anhMur*N*Ac and the peptide, and the experiment in Figure 2–figure supplement 2 shows that it does not cleave the bond between Mur*NAc* and peptides in intact peptidoglycan.

(b) Caption line 1439: with AmiR OR the catalytically...

Done.

(3) Figure 3Panel A: Label the products as NagZ-treated.

In this analysis, we quantify specific intermediates from the total cellular pool of PG recycling inter­mediates. Since the products were not specifically treated with NagZ, we would prefer to keep the figures as it is.

(4) Figure 4 (and Fig. 4-Figure Supplement 1, 2)(a) Please add complemented strains for ΔampG, ΔamiR, and ΔnagZ under the same conditions.

As described in more detail above, we have now performed the requested complementation analysis.

(b) Figure 4 - Figure S1 - Please include images of all strains quantified in B (e.g. control WT).

Done.

(c) Figure 4 - Figure S2: A. Please include images of all strains quantified in B. Please include spotting dilutions on minimal medium to assess the importance of PG recycling under nutrient limitation, especially given apparent lysis in ΔamiR and ΔampG.

The length distributions of cells grown in PYE medium are taken from Figure 3 and only shown for comparison (as mentioned in the figure legend). To avoid the duplication of images, we would prefer to keep panel A as it is.

We have now performed the requested serial-dilution spot assay on minimal (M2G) medium. The results show that ampicillin resistance de­creases even more dramatically for all strains in this condi­tion. The new data are presented in Figure 4–figure supplement 3C.

(d) Figure 4 - Figures S3: A and B. Please include WT control.

We have now added images of the wild-type strain to panel B of this figure. The serial dilution spot assays shown in panel A were performed on the same plates as those depicted in Figure 4 (as men­tioned in the figure legend). To avoid the duplication of images, we would prefer to keep this panel as it is.

(5) Figure 5A, C - please include images of WT control.

We have now added images of the wild-type strain to panel A of this figure. The serial dilution spot assays shown in panel C were performed on the same plates as those depicted in Figure 4 (as men­tioned in the figure legend). To avoid the duplication of images, we would prefer to keep this panel as it is.

(6) Figure 6:(a) A, C - please include images of WT control.

We have now added images of the wild-type strain to panel A of this figure. The serial dilution spot assays shown in panel C were performed on the same plates as those depicted in Figure 4 (as men­tioned in the figure legend). To avoid the duplication of images, we would prefer to keep this panel as it is.

(b) It would be informative to test ΔamgK and ΔanmK on minimal medium (spotting and/or growth curves) to position these steps within the nutrient-dependent fitness landscape.

We have now analyzed the ampicillin sensitivity of the Δ*amgK*, Δ*nagK* and Δ*amgK* Δ*nagK* strains on minimal medium (see Author response image 1). Consistent with the results obtained for other mutants in the PG recycling pathway, growth on minimal (M2G) medium plates leads to increased ampicillin sensi­tivity of the Δ*amgK* mutant. By contrast, Δ*nagK* and, to a lesser extent, Δ*amgK* Δ*nagK* cells show an in­creased tolerance to ampicillin under these conditions compared to growth on PYE plates.

This phenomenon may be explained by the strong stimulatory effect of Glc*N*Ac-6-P on NagB acti­vity. In the absence of NagK, Glc*N*Ac-6-P levels drop, leading to reduced activation of NagB1/2. This effect, combined with abundant glucose to support central carbon metabolism may promote the GlcN-6-P biosynthesis through GlmS, thereby increasing the flux of meta­bol­ites into the de novo PG biosynthesis pathway and thus boosting ampicillin tolerance. However, more re­search is required to fully under­stand the molecular basis of this effect. Given that the results are likely to reflect complex interactions bet­ween dysregulated enzyme activity and altered metabolite pools caused by increased glucose avail­ability, they provide only limited insight into the role of PG recycling in ampicillin resistance. We therefore propose excluding this experiment from the present manuscript to avoid confusion.

**Author response image 1. sa2fig1:** Serial-dilution spot assay investigating the ampicillin resistance of the indicated mutant strains on minimal (M2G) medium plates.

(c) Could Figures 6 and 7 be combined for better comparison and since there is no WT control? If so, could you also include the MurNAc cytoplasmic level quantification for the double mutant (Figure 7)?

We would prefer to keep the two figures separated to avoid creating an overly large figure that contains a total of nine panels. However, we have now included an additional panel in Figure 7 show­ing the levels of Mur*N*Ac in the double mutant.

(7) Figure 7. A, CPlease include images of WT control.

We have now added images of the wild-type strain (now panel B). The serial dilution spot assays (now panel D) were performed on the same plates as those depicted in Figure 4 (as men­tioned in the figure legend). To avoid the duplication of images, we would prefer to keep this panel as it is.

(8) Figure 8-S1D, FPlease include images of WT control.

Panel F of this figure already contains a wild-type control.

(9) Figure 10 A, CPlease include images of WT control and ∆amiR (A).

Done.

(10) Figure 11Consider adding or highlighting in this figure (in a simplified manner) the major PG recycling differences in Caulobacter? The current model doesn't really show any difference that is unknown.

This figure presents a model of the mechanism underlying the increased β-lactam sensitivity of PG recycling-deficient cells. Since the PG recycling pathway of *C. crescentus* is already presented in detail in Figure 1, we would like to keep this figure simple and thus leave it as it is.

(11) Comments by lines:(a) Line 192: Clarify that NagZ is also part of the rate-limiting step since there is no difference between AmiR or NagZ order of hydrolysis?

We have now omitted the statement that AmiR catalyzes the rate-limiting step in the PG recycling process, because our data do not allow definitive conclusions on this point.

(b) Line 201: Define "considerable fraction" since this is known, please and cite original reference(s).

Done.

(c) Line 203: Please also cite the primary papers where they have found that disruption of the PG recycling pathway in *E. coli* and *P. aeruginosa* doesn't result in morphological defects.

Since there are a number of papers that report PG recycling-deficient mutants of *E. coli* and *P. aeru­ginosa*, we would like to keep citing reviews to support this statement. However, we have now addi­tionally included a review by Park & Uehara (2008), which provides a detailed overview of PG recycling in bacteria.

(d) Line 220-223: Though there are no obvious morphological defects, several mutants (e.g., ΔamiR, ΔampG) appear to be lysing or stressed under minimal conditions. Could you include spotting assays and/or growth curves on minimal medium (Figure 4, Figure S2) to quantify fitness under nutrient limitation?

Have performed the requested serial dilution spot assays on minimal (M2G) medium plates and now present the data obtained in Figure 4–figure supplement 3C.

(e) Line 224: PG recycling has been found to contribute to the regulation of B-lactam resistance in several organisms, not just those two. Perhaps add "including *C. freundii* and *P. aeruginosa*"

Done.

(12) Typographical errors:(a) Line 284: "caron" should be carbon.

Done.

(b) Line 323: "Figure C" needs a figure number.

Done.

(c) Line 33: "regulaton" should be regulation.

Done.

**Reviewer #2 (Recommendations for the authors):**
(1) The study is well conducted and describes a number of experiments that significantly deepen previous findings. The conclusions of this paper are mostly well supported by data, but some experiments and data analysis may need to be clarified and extended.

Thank you for this positive evaluation.

(2) The data presented in Figures 2B and 2C show activities of AmiR and NagZ using LTase-cleaved cell wall preparations. Unfortunately, the preparations tested with the two enzymes should be identical, but apparently are not. Why aren't identical preparations used?

We are sorry for the confusion. As stated in the Methods section (page 28, lines 757 and 773), the AmiR activity assays used LT products from PG sacculi isolated from *E. coli* D456, whereas the NagZ activity assays used LT-products from PG sacculi isolated from *E. coli* CS703-1. Both strains have a higher penta­peptide content than wild-type *E. coli* D456 lacks PBPs 4, 5 and 6 and has a moderate level of pentapeptides. CS703-1 lacks PBPs 1a, 4, 5, 6, 7 as well as AmpC and AmpH, and is known to have a higher pentapeptide content than D456. These differences are the reason for the distinct muro­peptide profiles in panel B and C of Figure 2.

(3) I am missing a control experiment where muropeptides treated with NagZ were further digested with AmiR? This would show whether AmiR is able or not to cleave MurNAc-peptides. This is not evident from the provided experiments.

We have now tested the activity of AmiR towards anhMur*N*Ac-tetrapeptide in vitro. The results show that AmiR efficiently cleaves this Glc*N*Ac*-*free anhydro-muropeptide species, verifying that it can also act on turnover products that have been previously processed by NagZ. The new data are shown in Figure 2–figure supplement 5.

(4) The claim that PG recycling is critical, particularly upon transition to the stationary phase and under nutrient limitation, is not justified. It conflicts with the obvious morphological effects also in the exponential phase and with the absence of morphological defects in minimal medium: pronounced defects in rich PYE medium (Figure 4A/B) disappear in minimal M2G medium (Figure 4_figure supplement 2). It seems that catabolite repression effects apply here. Is the morphological effect in rich PYE medium reversed by adding glucose?

We agree that PG recycling is not considerably more important in stationary phase and have removed this statement. Interestingly, while PG recycling-deficient mutants show no obvious mor­phol­ogical defects in minimal (M2G) medium, their ampicillin sensitivity even increases under this condi­tion (new Figure 4–figure supplement 3C), confirming that morphological and resistance defects are not strictly coupled. Preliminary data indicate that the morphological defects of the mutant cells are also abolished upon growth in PYE+glucose medium. High glucose availability may promote increased de novo synthesis of PG precursors, thereby partially restoring the PG precursor pool. We propose that the morphological and resistance phenotypes develop at different degrees of PG precursor depletion. However, future research is required to clarify the precise molecular basis of this phenomenon.

(5) Figure 4: Why is the contribution of AmpG to ampicillin resistance much lower than for amiR or nagZ, despite ampG mutants showing the largest morphological defects? Does the accumulation of UDP-MurNAc or UDP-MurNAc-peptide correlate with ampicillin resistance, whereas the morphological effects correlate with the lack of precursors?

The exact reason why the Δ*ampG* mutant shows such a strong discrepancy in the severity of its morphol­ogical and resistance defects compared to the Δ*amiR* and Δ*nagZ* mutants remains unclear, because all of these deletions completely block the recycling of anhydro-muropeptides. The major difference in the Δ*ampG* mutant is its inability to import anhydro-muropeptides, causing their accu­mu­lation in the periplasm. We propose that periplasmic anhydro-muropeptides, in particular the penta­peptide-containing species, can interact with the substrate-binding sites of PG metabolic enzymes, thereby interfering with proper PG biosyn­thesis. Conversely, by interacting with transpep­tidases, they may reduce their accessibility to ampicillin and thus preserve their acti­vity under β-lactam stress, particularly under conditions in which low PG precursor availability reduces binding site occupancy and thus facilitates antibiotic association.